# Paracrine effect of regulatory T cells promotes cardiomyocyte proliferation during pregnancy and after myocardial infarction

Serena Zacchigna[1,2], Valentina Martinelli[3], Silvia Moimas [2,3], Andrea Colliva[1], Marco Anzini[2], Andrea Nordio[2], Alessia Costa[1], Michael Rehman[1], Simone Vodret[1], Cristina Pierro[1], Giulia Colussi [3], Lorena Zentilin[3], Maria Ines Gutierrez[3], Ellen Dirkx[3], Carlin Long[3], Gianfranco Sinagra[2], David Klatzmann[4,5] & Mauro Giacca [2,3]

Cardiomyocyte proliferation stops at birth when the heart is no longer exposed to maternal blood and, likewise, to regulatory T cells (Tregs) that are expanded to promote maternal tolerance towards the fetus. Here, we report a role of Tregs in promoting cardiomyocyte proliferation. Treg-conditioned medium promotes cardiomyocyte proliferation, similar to the serum from pregnant animals. Proliferative cardiomyocytes are detected in the heart of pregnant mothers, and Treg depletion during pregnancy decreases both maternal and fetal cardiomyocyte proliferation. Treg depletion after myocardial infarction results in depressed cardiac function, massive inflammation, and scarce collagen deposition. In contrast, Treg injection reduces infarct size, preserves contractility, and increases the number of proliferating cardiomyocytes. The overexpression of six factors secreted by Tregs (Cst7, Tnfsf11, Il33, Fgl2, Matn2, and Igf2) reproduces the therapeutic effect. In conclusion, Tregs promote fetal and maternal cardiomyocyte proliferation in a paracrine manner and improve the outcome of myocardial infarction.

[1] Cardiovascular Biology Laboratory, International Centre for Genetic Engineering and Biotechnology (ICGEB), 34149 Trieste, Italy. [2] Department of Medical, Surgical and Health Sciences, University of Trieste, 34127 Trieste, Italy and Center for Translational Cardiology, Azienda Sanitaria Universitaria Integrata di Trieste, 34129 Trieste, Italy. [3] Molecular Medicine Laboratory, International Centre for Genetic Engineering and Biotechnology (ICGEB), 34149 Trieste, Italy. [4] Sorbonne Université, UPMC Univ Paris 06, INSERM, UMR_S 959, Immunology-Immunopathology-Immunotherapy (i3), F-75005 Paris, France. [5] AP-HP, Groupe Hospitalier Pitié-Salpêtrière, Department of Biotherapies, Clinical Investigation Center in Biotherapy and Inflammation-Immunopathology-Biotherapy Department (DHU i2B), F-75013 Paris, France. These Authors contributed equally: Serena Zacchigna, Valentina Martinelli, Silvia Moimas. Correspondence and requests for materials should be addressed to S.Z. (email: zacchign@icgeb.org)

A major, still unresolved issue in the cardiac regeneration field is the reason why the proliferative capacity of cardiomyocytes (CMs) undergoes a grinding halt early after birth[1]. Besides major hemodynamic and biochemical events occurring after birth, one major change is a sudden lack of exposure to the maternal circulation, suggesting that circulating cells or serum factors might be involved in the exit of CMs from the cell cycle. In particular, regulatory T cells (Tregs) could play a role in this process, as they are expanded in the mother to promote maternal immune tolerance toward the fetus[2]. Tregs are $CD4^+$ T cells expressing the transcription factor Forkhead box protein 3 (FOXP3), in addition to specific anti-inflammatory cytokines, such as interleukin-10 (IL-10) and transforming growth factor-$\beta$ (TGF-$\beta$), which dampen an excessive effector immune response[3]. At the onset of pregnancy, a profound modulation of the maternal immune response occurs, which entails $CD4^+$ $CD25^+$ Treg cell activation and expression of variety of molecules (TGF-$\beta$, IL-10, IL-8, and IL-2 receptor, among others), which will blunt the immune response of the mother and license development of the semi-allogeneic fetus[4]. In addition to immune suppressive function, recent reports have revealed that Tregs also control non-immunological processes, including visceral adipose metabolism[5,6] and muscle repair[7]. Moreover, an involvement of Tregs in the susceptibility and outcome of ischemic heart disease was described in both experimental models and in humans[8–10].

More recently, a beneficial effect of Tregs in improving the healing after myocardial infarction by modulating monocyte/macrophage polarization was reported[11].

Myocardial repair following ischemic injury involves a series of inflammatory events. A transient recruitment of circulating neutrophils and monocytes is followed by an intense macrophage infiltration. Within days, macrophages progressively shift from a pro-inflammatory, M1-type phenotype, required for the clearance of necrotic and apoptotic debris, to an anti-inflammatory, M2-type phenotype, exhibiting various pro-regenerative functions, such as matrix remodeling and promotion of angiogenesis[12]. Tregs have been shown to induce M2-type macrophage polarization within the healing myocardium, associated with myofibroblast activation and increased expression of monocyte/macrophage-derived proteins, which foster wound healing[11].

Here, we explore the possibility that Tregs directly regulate CM proliferation and might thus be used to stimulate myocardial repair after an acute ischemic damage. Our results show that the medium conditioned by Tregs promotes CM proliferation, similar to the serum from pregnant animals. Proliferative CMs are detected in the heart of pregnant mothers and Treg depletion during pregnancy decreases both maternal and fetal cardiomyocyte proliferations. After myocardial infarction, Treg depletion results in depressed cardiac function, massive infiltration of inflammatory cells and scarce collagen deposition into the scar. In contrast, Treg injection reduces infarct size, preserves contractility and increases the number of proliferating CMs. The overexpression of six factors secreted by Tregs (Cst7, Tnfsf11, Il33, Fgl2, Matn2, and Igf2) reproduces their therapeutic effect. Thus, Tregs promote fetal and maternal CM proliferations in a paracrine manner and improve the outcome of myocardial infarction.

## Results

### Factors secreted by Tregs promote cardiomyocyte proliferation.
We collected serum from mice at post-natal day 0 (neonatal serum, NS), 2 months old adult mice (adult serum, AS), and pregnant female mice at E15, namely 2 weeks after the detection of the vaginal plug marking successful mating (pregnant serum, PS). The sera were added to primary cultures of neonatal rat ventricular CMs, containing >90% CMs, together with 5-ethynyl29-deoxyuridine (EdU), a uridine analog that is incorporated into newly synthesized DNA. After 2 days, the cells were stained for sarcomeric α-actinin to distinguish CMs and for EdU incorporation. The serum from pregnant animals significantly stimulated CM proliferation when compared to CMs treated with neonatal or adult sera (Fig. 1a, b).

To start assessing the possible contribution of Tregs to this mitogenic activity, we transiently depleted the Treg population in pregnant CD1 animals by injecting an anti-CD25 antibody (PC61) on days E10 and E15, as indicated in Fig. 1c ($n = 4$ per group). Treg depletion in pregnant animals is known to interfere with maternal tolerance and to increase resorption rate during the implantation phase and early pregnancy, but not in the late stage of allogeneic pregnancy[13]. Therefore, we started depleting Tregs in pregnant mice starting from E10, in order to minimally interfere with embryonic viability. Efficacy of depletion was assessed by flow cytometry by quantifying the number of $CD25^+$ cells in lymph nodes. As shown in Supplementary Fig. 1a, c, treatment with PC61 antibody eliminated $CD25^+$ cells; the depletion was sustained for 7 days. Of notice, the serum from pregnant animals upon Treg depletion (TDPS in Fig. 1a, b) resulted in the complete loss of the pro-proliferative activity on neonatal rat ventricular CMs.

We then determined the effect of Treg depletion in the heart of developing embryos, by either using the PC61 antibody or administering diphtheria toxin to DEREG mice, which carry a diphtheria toxin receptor (DTR)-eGFP transgene under the control of the Foxp3 promoter, thereby allowing specific depletion of Tregs by application of diphtheria toxin (DT) ($n = 4$ per group). Also in this case, the depletion of $eGFP^+CD25^+$ cells was evident a few days after toxin injection and lasted for about 1 week (Supplementary Fig. 1b, c). No increase in either the expression levels of pro-inflammatory molecules or the number of $CD45^+$ leukocytes was observed in the TD embryonic hearts compared to controls, ruling out any major autoimmune response upon Treg depletion (Supplementary Fig. 1b, e). Starting from day E11, the same animals were administered EdU intraperitoneally every 24 h until day E18, when CM proliferation was assessed in the embryonic heart ($n > 10$ per group; Fig. 1c). Analysis of EdU incorporation revealed a marked reduction in the number of $EdU^+$/α-actinin$^+$ CMs in depleted animals in both models of Treg depletion (Fig. 1d–f). A detailed analysis of CM proliferation rate was performed on atria, septum, right and left ventricle of the fetal heart, using both α-actinin and PCM1 antibodies to label CMs. In our experience, the combined use of these two markers allows for a reliable identification of CMs, since the use of α-actinin alone can bias the assignment of a proliferative nucleus to a CM, while the use of PCM1 alone can instead mislead identification, as several neonatal cardiac fibroblasts score positive, while some adult CM nuclei score negative for this marker (Supplementary Fig. 2). Comparable results were obtained using both CM-specific markers and confirmed a significant reduction in the number of $EdU^+$/α-actinin$^+$ and $EdU^+$/PCM1$^+$ CMs in all cardiac regions (Fig. 1f, left and right panels, respectively, $P < 0.05$, as determined by one-way analysis of variance and Bonferroni post hoc test). No major differences were observed in the proliferation rate between the four cardiac compartments.

To obtain further evidence that the pro-proliferative effect exerted by the serum of pregnant animals in vitro and in vivo was dependent on molecules secreted by Tregs, we collected the medium conditioned by Tregs cultured on CD3-coated plates in the presence of IL-2[14], and added it to cultures of neonatal rat

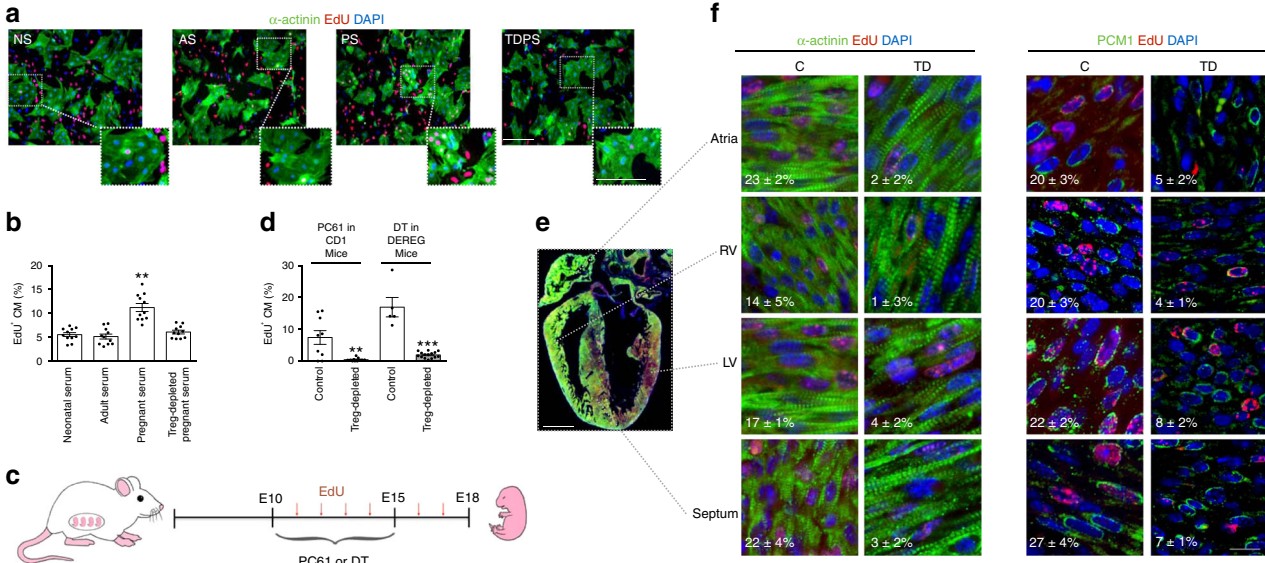

**Fig. 1** Tregs in the maternal serum promote CM proliferation. **a** Rat neonatal CMs were exposed to the serum collected from neonatal (NS), adult (AS), pregnant (PS) and Treg-depleted pregnant (TDPS) animals and proliferation measured by EdU incorporation. Nuclei are stained blue with 4'-6-diamidino-2-phenylindole (DAPI) and CMs green by anti-α-actinin antibodies. Red nuclei indicate EdU incorporation. Scale bar, 100 μm. Insets shows a higher magnification of the area defined by the white dotted line. **b** Quantification of EdU⁺ CM nuclei (% of total CMs) after exposure to the different sera. **c** Outline of the experimental procedure followed for Treg depletion in vivo in pregnant animal. The Treg-depleting agent, either PC61 antibodies in CD1 mice or Dyphteria Toxin (DT) in DEREG mice, was administered i.p. to the pregnant mother at embryonic days 10 (E10) and 15 (E15). EdU was administered i.p. every day from E10 to E18, when embryos were removed for histological analysis. **d** Quantification of EdU⁺/ α-actinin CM nuclei (% of total CMs) in the heart of embryos harvested from either control (white bars) or Treg-depleted (gray bars) mothers, in the two models of Treg depletion described in panel (**c**). **e** Longitudinal section of a whole E18 heart, in which nuclei are stained blue with DAPI, CMs green by anti-α-actinin antibodies and EdU incorporation is shown in red. Scale bar, 1 mm. **f** High magnification images of the indicated regions of embryonic hearts from control (C) and Treg-depleted (TD) mothers, in which nuclei are stained blue with DAPI, CMs green by either anti-α-actinin or PCM1 antibodies and EdU incorporation is shown in red. RV: right ventricle, LV: left ventricle. White numbers indicate the percentage of EdU⁺ CMs in each cardiac region. Scale bar, 20 μm. Values in (**b**) and (**d**) are mean ± s.e.m., $n \geq 3$ biological replicates. One-way analysis of variance and Bonferroni/Dunn's post hoc tests were used to compare multiple groups (**b**). Pairwise comparison was performed with the Student's t-test (**d**). **$P < 0.01$, ***$P < 0.001$ relative to control

ventricular CMs. These primary cultures also contain a few non-CM cells, of which >99% are fibroblasts[15]. Cell proliferation was assessed by EdU incorporation, as well as by staining for Ki67 and histone H3 phosphorylated on serine10 (H3PS10), which represent markers of the all active phases of the cell cycle and of late G2/mitosis, respectively. Exposure to the Treg-condition medium significantly increased the percentage of all the three proliferation markers in CMs but not in fibroblasts (Fig. 2a–d). To demonstrate that the increase in CM DNA replication eventually resulted in cell division, we stained cells for Aurora B kinase. Treatment with Treg-conditioned medium significantly increased the number of cells presenting Aurora B localized at midbodies, which are transient structures formed during cytokinesis (Fig. 2a, e).

Taking these data collectively, it can be concluded that Tregs secrete soluble factors that promote embryonic and neonatal CM proliferation in vitro and in vivo.

**Tregs contribute to increased heart size during pregnancy.** Inspired by the evidence that Tregs exert a pro-proliferative effect on embryonic and neonatal CMs, we wondered whether a similar activity might occur in the heart of pregnant animals, in which the Treg pool is expanded to secure maternal tolerance toward fetal antigens. It is well known that the heart experiences an increase in size during pregnancy. This phenomenon has been traditionally ascribed to a reversible, compensatory enlargement in CM size in response to the increased circulating volume occurring in pregnant organisms and therefore considered an example of transient physiological hypertrophy. We

hypothesized that an increase in CM number, in addition to their size, might contribute to the gross enlargement of the heart during pregnancy.

First, we confirmed that both heart size and the percentage of circulating Tregs increased during pregnancy; while the heart progressively enlarged during the whole duration of pregnancy until delivery, the percentage of Tregs peaked at E12 and then decreased at subsequent times ($n = 5$; Fig. 3a–c). CM cross-sectional area was also significantly increased at delivery (Fig. 3d), despite pregnancy is known to increase CM length more than their diameter[16].

We then assessed CM proliferation by counting the number of EdU⁺ CMs in the hearts of pregnant animals, repeatedly injected with EdU starting from E10, as described in Fig. 1c. We found that the number of EdU⁺ CMs in the mothers' hearts was increased during pregnancy and at the time of delivery; in particular, proliferation peaked at E12 (Fig. 3e, f and Supplementary Fig. 3a), at the time when Tregs number was the highest in the maternal circulation (Fig. 3c). A few CMs in E12 hearts showed positivity for the proliferation marker Ki67 (Supplementary Fig. 3b) and AuroraB localization at midbodies, which indicated passage through mitosis and was never observed in control CMs (Fig. 3g, h). Of notice, the number of proliferating CMs, detected by both EdU incorporation and AuroraB staining was reduced in the heart of Treg-depleted, pregnant animals (Supplementary Fig. 3c, d).

Thus, CM proliferation occurs in the heart of pregnant animals, at least in part sustained by Tregs, contributing to the gross increase in heart size during pregnancy.

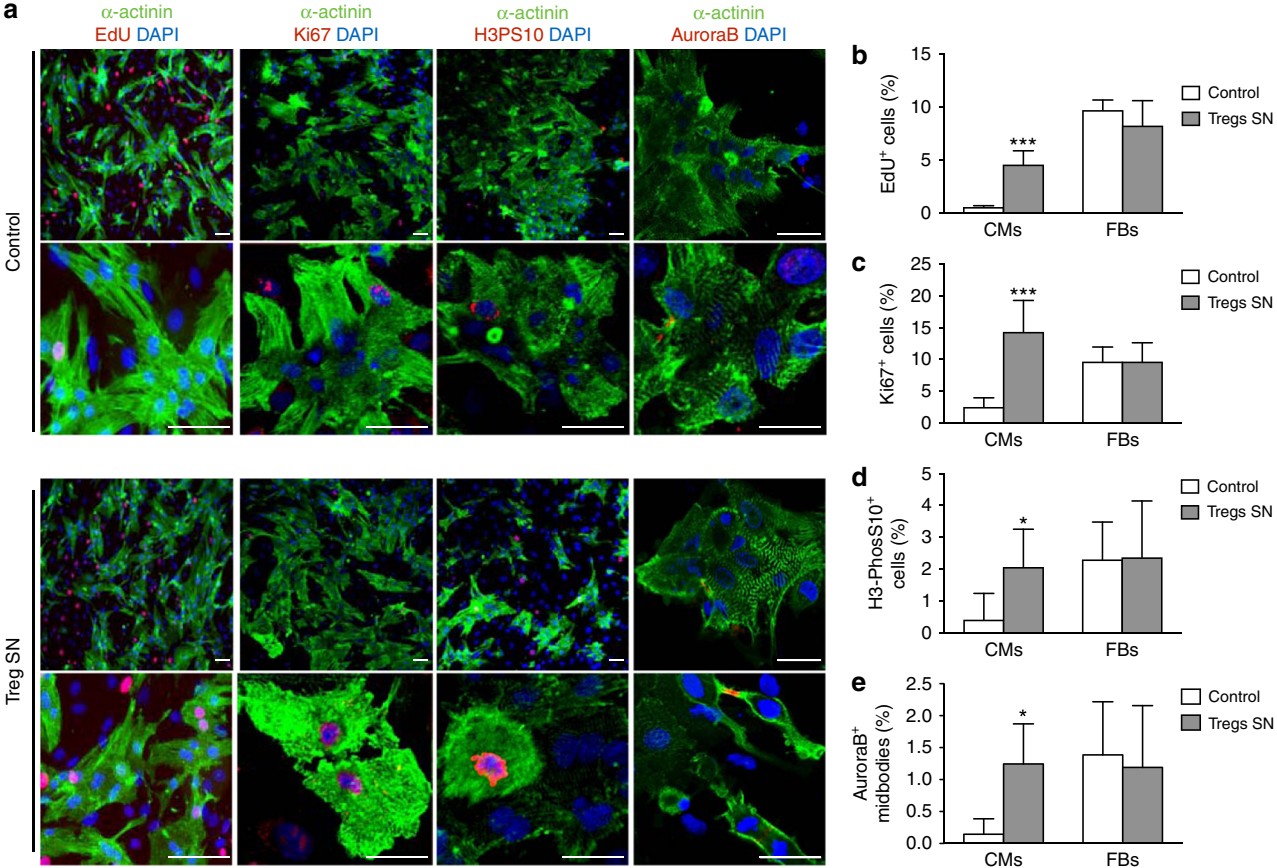

**Fig. 2** Tregs promote the proliferation of neonatal CMs. **a** Rat neonatal CMs were exposed to the supernatant (SN) conditioned by Tregs and stained for various proliferation markers, including EdU incorporation, Ki67, histone H3 phosphorylated on serine10 (H3PS10), and Aurora B. All proliferation markers are stained red, while nuclei are stained blue with DAPI and CMs green using anti-α-actinin antibodies. Control cells were exposed to the medium normally used to culture Tregs, containing recombinant IL2 (50 ng/ml). Scale bar, 25 μm. **b–e**. Quantification of EdU+ (**b**), Ki67+ (**c**), H3PS10+ (**d**) cells, and AuroraB+ localization in midbodies (**e**) in either CMs (CMs, white bars) or fibroblasts (FBs, gray bars), expressed as the percentage of the total number of nuclei belonging to each cell type. All values are mean ± s.e.m., $n = 3$ biological replicates. At least 500 cells were analyzed per each replicate using the MetaXpress software. Pairwise comparison was performed with the Student's t-test (**b–e**); *$P < 0.05$, **$P < 0.01$, ***$P < 0.001$ relative to control

**Endogenous Tregs protect the heart after myocardial infarction.** Since the previous results were consistent with effect of Tregs in inducing CM proliferation in both developing and adult hearts, we explored whether Tregs might also exert a beneficial, pro-regenerative role in adulthood following myocardial infarction (MI). By exploiting the presence of eGFP+ Tregs in untreated DEREG mice, we analyzed the endogenous recruitment of Tregs at different time points after the induction of MI by permanent ligation of the left anterior descending coronary artery ($n = 16$ per group; 4 animals per time point). A significant number of Tregs was detected in the region of the heart surrounding the ischemic zone, already a few hours after coronary artery ligation; these cells persisted in the infarcted area for at least 7 days (Supplementary Fig. 4a, b). Treg recruitment at the site of MI was also confirmed by the increased level of Foxp3 mRNA at the same time points (Supplementary Fig. 4c). In line with the presence of Tregs in the ischemic heart, the expression of TGF-β and IL-10, two factors abundantly produced by Tregs, was significantly increased during the first week after coronary artery ligation and, particularly, a few hours after the ischemic insult (Supplementary Fig. 4d).

We then assessed the outcome of MI in CD1 mice that had been depleted of Tregs using an anti-CD25 antibody 5 days in advance of and 30 days after surgery ($n = 12$ per group). Prolonged Treg depletion increased the number of major lethal events, such as apical aneurysms and cardiac ruptures, resulting in a net increase in mortality in the Treg-depleted group compared to not depleted animals (Fig. 4a, b). As evaluated by ultrasound imaging in surviving mice, left ventricular ejection fraction (LVEF, Fig. 4c), fractional shortening, end-systolic anterior wall thickness, thickening, and internal diameter (Supplementary Fig. 5), were mildly but consistently more compromised over time in Treg-depleted mice. Morphometric analysis of the infarcted hearts confirmed the presence of larger infarcts in Treg-depleted animals (Fig. 4d, e). At histological examination, Treg depletion resulted in a massive accumulation of CD45+ inflammatory cells (Fig. 4f, g), which could have determined looser scars and propensity to rupture. No significant changes in the extent of cell death were detected in Treg-depleted hearts compared to controls (Supplementary Fig. 6).

To better investigate the effect of Tregs in the fibrotic response, we performed MI and Treg depletion in Colα1(I)-EGFP mice, in which EGFP expression is confined to fibroblasts[17]. Both the number of EGFP+ fibroblasts (Fig. 4h, i) and the intensity of EGFP expression, which is a surrogate indicator of collagen expression (Fig. 4h, j), were significantly reduced in Treg-depleted mice.

Collectively, these data are consistent with the conclusion that Tregs exert a protective role in MI healing by resolving

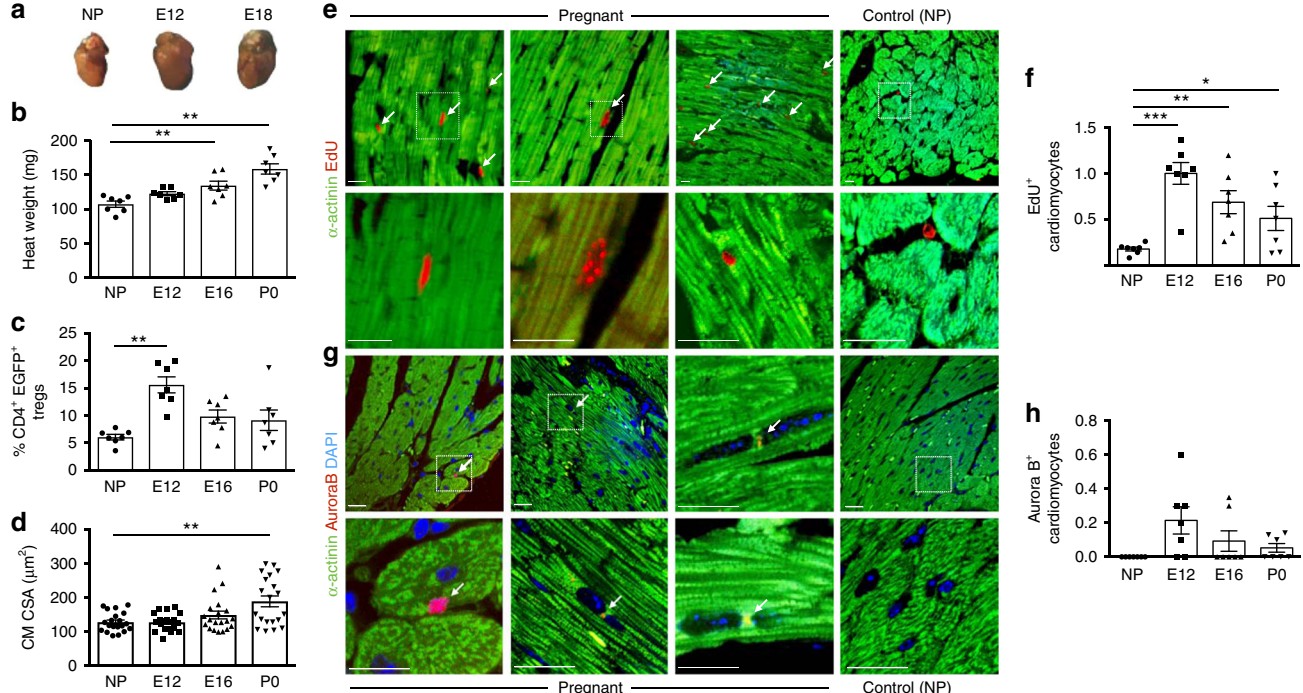

**Fig. 3** Hyperplasia contributes to increased heart size in pregnancy. **a** Hearts harvested from not pregnant (NP) and pregnant female mice, at the indicated stages of embryonic development (E12 and E18). **b** Quantification of weight of the heart in not pregnant (NP, circles) and pregnant female mice, at the indicated stages of embryonic development (E12, squares and E18, tip up triangles) and at the day of the delivery (P0, tip down triangles). **c** Quantification of circulating Tregs in the peripheral blood of DEREG mice, expressed as the percentage of EGFP+ cells over the total number of CD4+ cells at the same time points as in (**b**). **d** Quantification of the CM cross-sectional area (CSA) in heart sections of mice at different stages of pregnancy, as in (**b**, **c**). **e** Sections of the heart of pregnant mice at E12, showing a relatively high level of EdU incorporation (stained red) in the nuclei of CMs (stained green) compared to a control, not pregnant (NP) female heart. Scale bar, 100 μm. **f** Quantification of EdU incorporation (% of total CM nuclei) in CMs of not pregnant (NP) and pregnant female mice, at the indicated stages of embryonic development (E12 and E18) and at the day of the delivery (P0). **g** Sections of the heart of pregnant mice at E12 showing Aurora B localization in midbodies (stained red) of CMs (stained green by anti-α-actinin antibodies). Nuclei are stained blue with DAPI. No Aurora-B signal is present in control, not pregnant (NP) female heart. Scale bar, 100 μm. **h** Quantification of the AuroraB+ (% of total CM nuclei) in CMs of not pregnant (NP) and pregnant female mice, at the indicated stages of embryonic development (E12 and E18) and at the day of the delivery (P0). All values are mean ± s.e.m., each dot indicates a biological replicate. One-way analysis of variance and Bonferroni/Dunn's post hoc tests were used to compare multiple groups. *P < 0.05, **P < 0.01, ***P < 0.001 relative to NP

inflammation and improving collagen deposition, thereby protecting from cardiac rupture and preserving cardiac function.

**Treg injection at the site of MI promotes CM proliferation**. To better discriminate between the immuno-modulatory and the pro-proliferative activities of Tregs after MI, we assessed their effect in mouse models of both ischemia-reperfusion and permanent coronary artery ligation ($n = 8$ per group), which better reproduces the clinical condition and is associated to a different pattern of cytokine expression and leukocyte recruitment to the heart[12,18]. For both experimental approaches, we purified EGFP+ Tregs from DEREG mice by cell sorting (purity of sorting was 92% as shown in Supplementary Fig. 7). Immediately after isolation, we injected $1.5 \times 10^5$ cells in the peri-ischemic area of the heart of C57BL/6 syngeneic mice at the time of coronary artery ligation. After 40 min of ischemia, the artery was made patent again by surgical removal of the wire. The injected Treg cells engrafted into the recipient heart and were still present at day 3 after injection, as determined by direct EGFP localization and anti-EGFP immunostaining (Fig. 5a). Treg supplementation resulted in reduced infarct size (Fig. 5b), associated with improved cardiac function up to 3 months after MI (Fig. 5c).

We then assessed the effect of the same number of Tregs after permanent coronary artery ligation. Also in this case, Treg

supplementation was effective in preserving LVEF (Fig. 5d), as well as other parameters of cardiac function, such as end-systolic anterior wall thickness, thickening and internal diameter (Supplementary Fig. 8).

Based on the previous evidence that Tregs promote CM proliferation during embryonic development and early post-natal life, we scored the presence of proliferating CMs in a subgroup of animals that underwent MI, followed by Treg injection and EdU administration every other day for 2 weeks. The number of EdU+ CMs in the peri-infarct region was almost negligible in control animals and significantly increased upon Treg supplementation (Fig. 5e–g). Of notice, we could detect a few Tregs still present in the heart at 2 weeks after injection, in close proximity to EdU+ CMs (Fig. 5h).

We also performed MI in genetically modified animals presenting various impairments in the immune function, such as athymic Nude-Foxn1 (lacking T cell activity), Fox Chase SCID (lacking T and B cells), and Fox Chase SCID BEIGE (presenting T and B cell deficiency and diminished NK cell activity) mice. In all these immunodeficient mice, we could not detect changes in the outcome of MI (Supplementary Fig. 9a–c) nor in increased EdU incorporation (Supplementary Fig. 9d), indicating that the immune system does not play a major role in this setting, and likewise indicating that the effect of Tregs in improving MI outcome is rather dependent on their capacity to

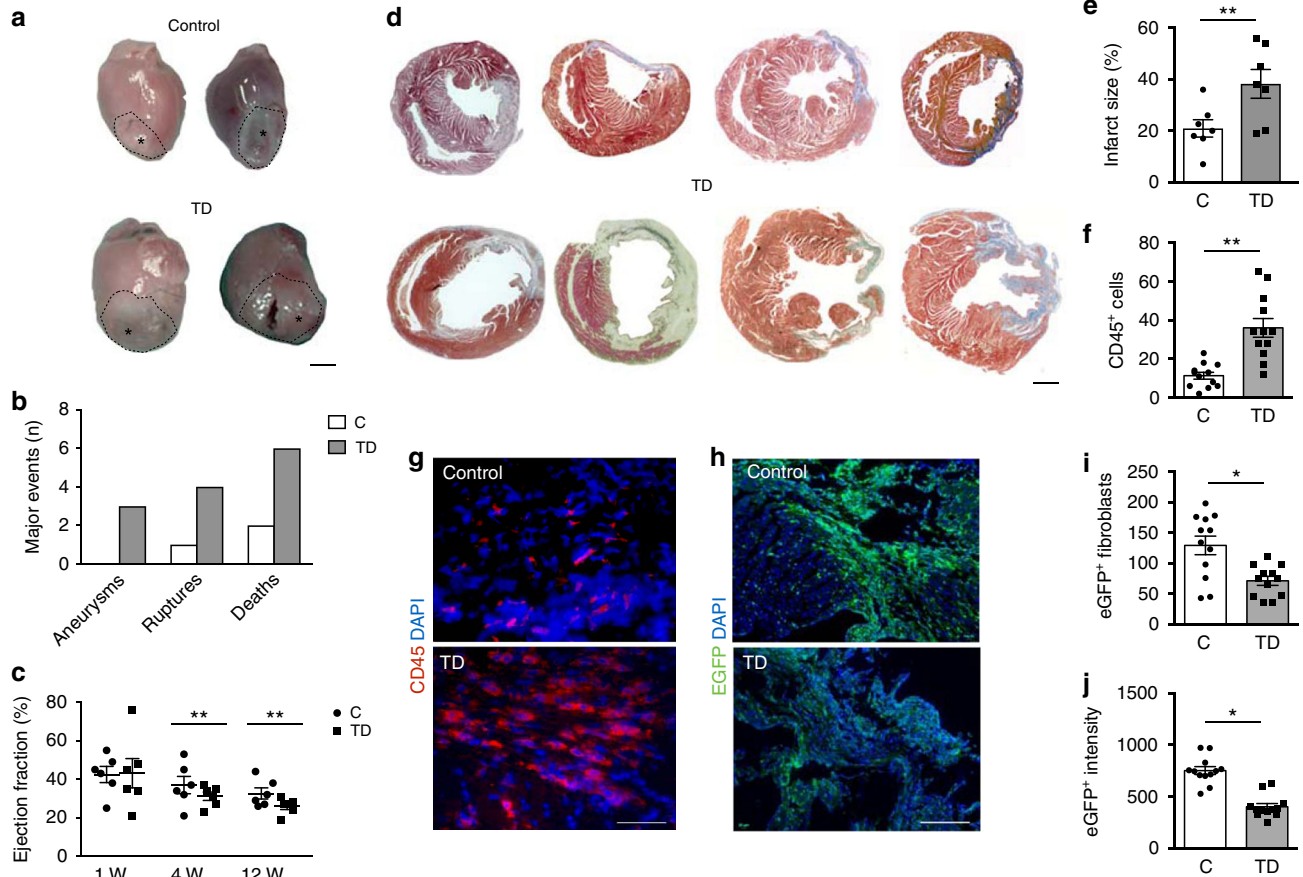

**Fig. 4** Treg depletion worsens the outcome of myocardial infarction. **a** Hearts at 1 month after the ligation of the left descendent coronary artery in CD1 control mice and in mice depleted of Tregs using an anti-PC61 antibody. Scale bar, 1.5 mm. **b** Quantification of the number of mice experiencing aneurysm formation, cardiac rupture or death during the follow-up of the study (3 months) in the control (C) and Treg-depleted (TD, gray bars) groups. **c** US imaging analysis of the left ventricular ejection fraction of control (C, circles) and Treg-depleted (TD, squares) mice at 1, 4, and 12 weeks (W) after myocardial infarction. **d** Representative images of whole transverse sections after Azan Trichromic staining of hearts of control (C) and Treg-depleted (TD) animals. Fibrotic areas are stained in gray/blue. Scale bar, 1 mm. **e** Quantification of infarct size (fibrotic area in D), expressed as percentage of the left ventricular area in control (C) and Treg-depleted (TD) mice at 3 months after myocardial infarction. **f** Quantification of CD45[+] cells in the scar region of control (C) and Treg-depleted (TD) mice (% of total nuclei per field) at 1 month after myocardial infarction. **g** Heart sections from control (C) and Treg-depleted (TD) animals stained by anti-CD45 pan leukocyte antibodies (red). Nuclei are stained blue with DAPI. Scale bar, 25 μm. **h** Representative images of heart sections at 1 month after myocardial infarction in control (C) and Treg-depleted (TD) Collα1(I)-EGFP mice, showing green fibroblasts and nuclei stained with DAPI. Scale bar, 100 μm. **i, j** Quantification of the number of EGFP[+] fibroblasts (**i**) and EGFP fluorescence intensity (**j**) in the scar region of control (C) and Treg-depleted (TD) Collα1(I)-EGFP mice (% of total nuclei per field) at 1 month after myocardial infarction. All values are mean ± s.e.m., each dot indicates a biological replicate; $n = 7$ biological replicates in (**b**). Pairwise comparison was performed with the Student's $t$-test (**e, f, i, j**). Two-way ANOVA for repeated measurements was used in (**c**). *$P < 0.05$, **$P < 0.01$, relative to control

induce CM proliferation rather than to modulate the immune response.

Thus, the injection of an extra-amount of Tregs at the site of cardiac ischemia exerts a beneficial effect on lesion healing, at least in part by sustaining CM proliferation and independent from their immunosuppressive function.

**Six secreted proteins surrogate the mitogenic effect of Tregs.** To identify putative Treg-secreted factors responsible for the observed proliferative effect, we performed an in silico analysis and compared the transcriptome of CD4[+]/CD25[+] Tregs with that of CD4[+]/CD25[−] lymphocytes from the GEO database (https://www.ncbi.nlm.nih.gov/geo/query/acc.cgi?acc=GSE4571). We selected 12 secreted factors, which were abundantly expressed by Tregs but not by CD4[+]/CD25[−] lymphocytes, of which an ORF clone was available and suitable for cloning into an AAV plasmid (Table 1). First, we tested the potential of each factor to

individually promote the proliferation of cultured CMs. To obtain an enriched source of each factor, we transfected CHO cells with each of the encoding plasmids and, after 48 h, we collected the serum-free conditioned medium. The 12 supernatants, together with a control medium, collected from CHO cells transfected with an empty plasmid (pGi), were added to neonatal rat CMs, followed by analysis of EdU incorporation. We identified six secreted factors that were individually able to increase CM proliferation, shown as black bars in Fig. 6a. These included Cst7, Tnfsf11, Il33, Fgl2, Matn2, and Igf2. Based on these results and to investigate potential synergy between the identified factors, we pooled either all the 12 supernatants (Pool12) or those corresponding to the six factors that were active individually (Pool6). We found that Pool6 was superior to any individual factor and that no further improvement was obtained by the inclusion of all 12 factors (Fig. 6a, b).

We moved on by testing the potential of the Treg-derived soluble factors to surrogate the protective effect exerted by the

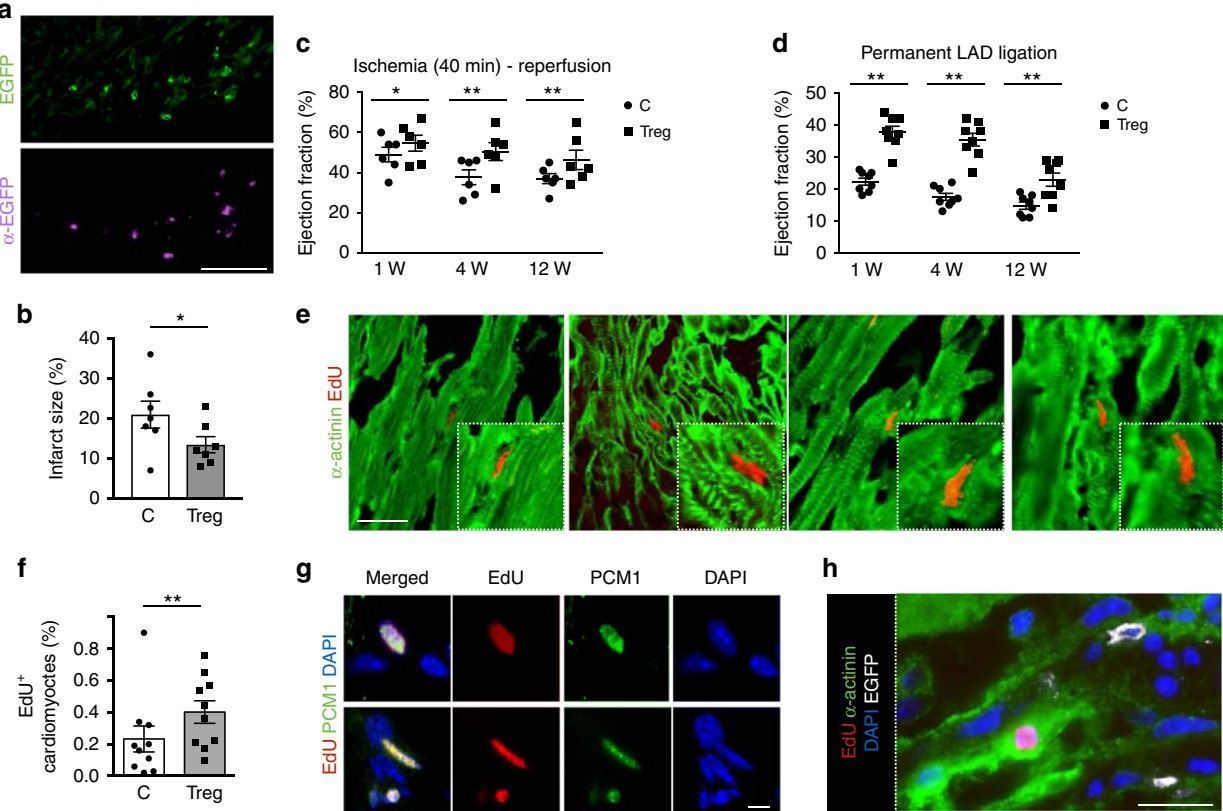

**Fig. 5** Treg administration improves the outcome of myocardial infarction. **a** Tregs sorted by DEREG mice were injected into the border region of the myocardial infarction and visualized as EGFP[+] cells in heart section at day 3 after surgery, detecting either the endogenous green fluorescence (upper panel) or using anti-EGFP antibodies (purple, lower panel). Scale bar, 25 µm. **b** Quantification of infarct size expressed as percentage of the left ventricular area in control (C, circles) and Treg-injected (Treg, squares) mice at 3 months after myocardial infarction. **c** US imaging analysis of the left ventricular ejection fraction of control (C) and Treg-injected (Treg) mice at 1, 4, and 12 weeks (W) after 40 min (min) of coronary artery ligation followed by vessel reperfusion. **d** US imaging analysis of the left ventricular ejection fraction of control (C) and Treg-injected (Treg) mice at 1, 4, and 12 weeks (W) after permanent ligation of the left descendent anterior (LAD) coronary artery. **e** Sections of the heart of Treg-injected mice, showing EdU incorporation (stained red) in the nuclei of CMs (stained green by anti-α-actinin antibodies). Scale bar, 100 µm. **f** Quantification of the EdU incorporation (% of total CM nuclei) in CMs in control (C) and Treg-injected (Treg) hearts at 15 days after permanent coronary artery ligation. **g** Sections of the heart of Treg-injected mice, showing EdU incorporation (stained red) in the nuclei of CMs (stained green by anti-PCM1 antibodies). Nuclei are stained in blue with DAPI. Scale bar, 10 µm. **h** Section of the heart showing injected Tregs (labeled in white by anti-EGFP antibodies) in proximity of one EdU[+] CM (EdU is labeled in red and CMs in green by anti-α-actinin antibodies). Nuclei are stained in blue with DAPI. Scale bar, 50 µm. All values are mean ± s.e.m. each dot indicates a biological replicate. Pairwise comparison was performed with the Student's *t*-test (**b**, **f**). Two-way ANOVA for repeated measurements was used in (**c**, **d**). *$P < 0.05$, **$P < 0.01$, relative to control

**Table 1 Secreted proteins highly expressed by Treg cells compared to CD4[+]/CD25[−] cells**

| Gene symbol | Gene name | Relative expression (Treg over CD25−) | Localization |
|---|---|---|---|
| Fgl2 | Fibrinogen-like protein 2 | 6.820291067 | Extracellular space |
| Matn2 | Matrilin 2 | 6.469789887 | Extracellular space |
| Tnfsf11 | Tumor necrosis factor (ligand) superfamily, member 11 | 5.290882688 | Extracellular space |
| Prss8 | Protease, serine, 8 (prostasin) | 4.66635417 | Extracellular space |
| Cst7 | Cystatin F (leukocystatin) | 4.397173444 | Extracellular space |
| Igfbp3 | Insulin-like growth factor binding protein 3 | 4.226069724 | Extracellular space |
| Il33 | Interleukin 33 | 4.18511797 | Extracellular space |
| Igf2 | Insulin-like growth factor 2 | 4.111991921 | Extracellular space |
| Cfhr1 | Complement factor H-related 1 | 3.921560832 | Extracellular space |
| Lta | Lymphotoxin A | 3.834880388 | Extracellular space |
| Hapln2 | Hyaluronan and proteoglycan link protein 2 | 3.771416801 | Extracellular space |
| C2 | Complement component 2 (within H-2S) | 3.660525412 | Extracellular space |

whole cells. We produced AAV9 vectors for the in vivo expression of the factors contained in both Pool6 and Pool12 and injected them in the peri-ischemic area of the heart of mice at the time of coronary artery ligation ($n = 8$ per group). An AAV9 vector not coding for any protein was used as a control. All mice received EdU throughout the duration of the experiment. Cardiac function was assessed by echocardiography over time, followed by morphometric and histological analysis. Both assessments

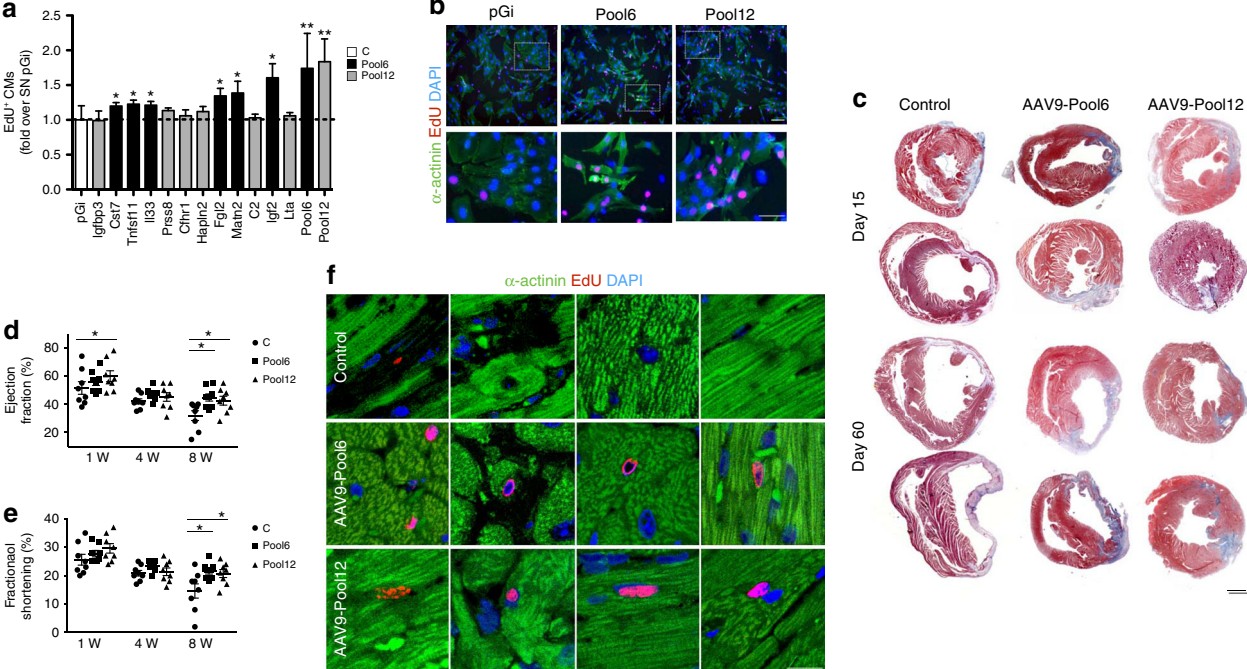

**Fig. 6** Tregs secrete a pool of proteins promoting CM proliferation. **a** Supernatants from CHO cells transfected with plasmids encoding for the factors listed on the X-axis of the graph were added to rat neonatal CMs and proliferation scored as EdU incorporation. All the tested proteins were included in Pool12, whereas only those exerting a pro-proliferative effect individually were included in Pool6. Black bars correspond to proteins included in Pool6, whereas gray bars correspond to proteins included in Pool12 but not in Pool6. The supernatant of cells transfected with an empty plasmid (SN pGi) was used as a negative control. At least 500 cells were analyzed per supernatant in each replicate using the MetaXpress software. **b** Representative pictures showing EdU incorporation by rat neonatal CMs exposed to the supernatant of CHO cells transfected with the negative control pGi, Pool6, and Pool12. Lower panels show higher magnification of the are defined by the white dotted line in the upper panels. Nuclei are stained blue with DAPI and CMs green by anti-α-actinin antibodies. Red nuclei indicate EdU incorporation. Scale bar, 25 μm. **c** Representative images of whole transverse sections after Azan Trichromic staining of hearts injected with an empty AAV vector (Control) or AAV vectors expressing the proteins included in Pool6 and Pool12 at 15 and 60 days after myocardial infarction. Fibrotic areas are stained in gray/blue. Scale bar, 1 mm. **d** US imaging analysis of the left ventricular ejection fraction in animals injected with a control vector (C) and vectors expressing the proteins included in Pool6 and Pool12 at 1, 4, and 8 weeks (W) after permanent coronary artery ligation. **e** US imaging analysis of the left ventricular fractional shortening in animals injected with a control vector (C) and vectors expressing the proteins included in Pool6 and Pool12 at 1, 4, and 8 weeks (W) after permanent coronary artery ligation. **f** Sections of the heart of mice injected with a control vector, AAV-Pool6, and AAV-Pool12, showing EdU incorporation (stained red) in the nuclei of CMs (stained green). Nuclei are stained blue with DAPI. Scale bar, 50 μm. All values are mean ± s.e.m., each dot indicates a biological replicate. One-way analysis of variance and Bonferroni/Dunn's post hoc tests were used to compare multiple groups in a. Two-way ANOVA for repeated measurements was used in (**d**). *$P < 0.05$, **$P < 0.01$, relative to control

indicated a significant reduction of infarct size at day 15 (22% and 35% reduction in the AAV9-Pool6 and AAV9-Pool12 groups, respectively; $P < 0.05$ in both cases, as determined by one-way analysis of variance and Bonferroni post hoc test). This reduction remained stable at day 60 (22% and 34% reduction in the AAV9-Pool6 and AAV9-Pool12 groups, respectively; $P < 0.05$ in both cases, as determined by one-way analysis of variance and Bonferroni post hoc test), as shown by a few representative hearts in Fig. 6c. These pathological observations correlated with significantly improved cardiac function in the animals treated with the two AAV pools. Left ventricle ejection fraction and fractional shortening were improved through the whole echocardiographic follow-up period, with the most beneficial effect evident at the latest time point (Fig. 6d, e). Analogous results were obtained analyzing other parameters of cardiac function, including left ventricle anterior wall systolic thickness (LVAWs) and left ventricle systolic internal diameter (LVID), the main indicator of cardiac dilation and dysfunction (Supplementary Fig. 10a, b).

Analysis of CM proliferation revealed a significant increase in the number of CMs undergoing DNA synthesis (number of EdU$^+$/α-actinin$^+$ cells: 0.2 ± 0.1% in control, 0.6 ± 0.3% in AAV9-Pool6, and 0.8 ± 0.3% in AAV9-Pool12 animals), as shown in

Fig. 6f. In AAV9-Pool12 we also detected the presence of PCM1$^+$ nuclei scoring positive for H3PS10, indicating cell cycle progression to late G2/mitosis (Supplementary Fig. 10c).

Thus, soluble factors secreted by Tregs reduce infarct size, preserve cardiac function and promote CM proliferation after myocardial infarction.

## Discussion

Collectively, the results shown here indicate that CD4$^+$Foxp3$^+$ Tregs act in a paracrine manner to promote CM proliferation during development, in both the maternal and the embryonic heart, and after myocardial infarction. They also show that a pool of key factors is responsible for this mitogenic activity in cultured cells and in vivo.

It is well established that an expansion of the Treg compartment occurs during pregnancy. In mice, a monthly increase in Tregs within the uterus during the estrus cycle prepares the uterus for a potential embryo implantation. If the implantation occurs, the expansion continues, first involving only thymus-derived Tregs and then also involving peripherally induced Tregs, reaching the highest level when trophoblast invasion to decidua is

maximal (2nd semester in human and E12–14 in mice)[19,20]. Although these events have been so far correlated to the regulation of the maternal immune response against feto-placental graft in the uterus, it is reasonable that these cells might affect other organs in both the fetal and the maternal body. Our results support this hypothesis as they show a direct effect of Tregs on CMs, independent of their immune modulatory function. Indeed, the supernatant of cultured Tregs was able to stimulate DNA proliferation and cytokinesis on primary CMs and Treg-depletion in vivo resulted in reduced CM proliferation in the developing embryonic heart. Of notice, we also showed that a certain rate of CM proliferation also occurs in the heart of pregnant mice, at least in part as a consequence of Treg expansion, as Treg-depletion in pregnant mice resulted in a reduced number of proliferating CMs. This reduction was not statistically significant, likely because Treg depletion was started at E10, in order to minimize the risk of resorption, and CM proliferation was assessed at its peak (E12), when the efficiency of depletion was still low. These data, however, open a window of novelty on the pathophysiology of the increased cardiac size always observed during pregnancy and traditionally called physiological hypertrophy to refer to the phenotypic gross similarities with pathological hypertrophy, which is the consequence of hypertension, myocardial infarction, valvular heart diseases, or genetic mutations. While physiological hypertrophy, which also occurs upon exercise, is characterized by normal or even enhanced contractile function coupled with normal architecture and cardiac structure, pathological hypertrophy associates with CM death and fibrotic remodeling, and is characterized with reduced systolic and diastolic function, often progressing toward heart failure. Among the triggering events of cardiac hypertrophy, mechanical stress and neurohormonal stimulation are considered of primary importance in both cases. In pathological conditions, the hypertrophy of the left ventricle appears to be beneficial in the short-term and detrimental in the long-term harm[21], however, the exact mechanisms regulating the transition from adaptive to maladaptive hypertrophy are yet to be determined. Our data indicate that Tregs are recruited to the heart during pregnancy and early after myocardial infarction, two conditions considered to provide compensatory benefit and preserve heart function. These data tend to suggest that in both instances they could contribute to enhance cardiac contractility not only through the induction of CM enlargement but also through the stimulation of their proliferation.

In agreement with this concept, CM cross-sectional area significantly increased at the end of the pregnancy, but was rather stable at the earliest time point (E12), when the heart weight appeared modestly but already increased. At the same time point, both Treg number in the peripheral blood and the number of EdU$^+$ CM reached their highest values. At the same time point, a few AuroraB$^+$ CMs could also be observed, suggesting that a round of CM hyperplasia, followed by hypertrophy, could be responsible for the increased cardiac size during pregnancy.

Other reports have previously shown a beneficial effect of Tregs on heart function, especially after myocardial infarction. This effect has been so far ascribed to the capacity of Tregs to favorably interfere with cardiac remodeling, by inhibiting the production of pro-inflammatory cytokines[10] and inducing an M2-like macrophage differentiation, associated with myofibroblast activation and expression of macrophage-derived proteins fostering myocardial healing[11]. Our data confirm an essential and beneficial role of Tregs after acute heart ischemia, as their depletion in vivo determined a net increase in infarct size, reasonably due to the increased number of inflammatory cells at the level of the scar, often resulting in aneurysm formation and cardiac rupture. At the same time, our findings indicate that Tregs might protect the heart through an additional, previously unnoticed mechanism, namely by secreting proteins able to stimulate CM proliferation.

Our data add to the now growing list of Treg functions, which extend far beyond immune regulation. For instance, Tregs located within the visceral adipose tissue control glucose uptake by cultured adipocytes[6], while in the skeletal muscle they enhance the colony-forming efficiency of satellite cells and promote muscle repair[7]. Of interest, Tregs act in a paracrine manner to potentiate muscle repair, essentially through the secretion of the growth factor Amphiregulin, which is able to induce the myogenic differentiation of culture satellite cells and also to restore the muscle-specific reparative capacity, otherwise lost in Treg-depleted mice[7]. Our results are in line with the capacity of Tregs to promote the regeneration of other cell types in a paracrine manner, as the proliferative activity of Tregs on both cultured CMs and in vivo appears to be largely mediated by a pool of secreted proteins, abundantly secreted by Tregs compared to CD4$^+$/CD25$^-$ T lymphocytes. While there is no direct evidence that maternal Tregs can cross the placenta, (i) they have been found in the fetal lymph nodes and (ii) they produce cytokines that can cross the placenta and reach the fetal heart. Among these, six secreted proteins (Cst7, Tnfsf11, Il33, Fgl2, Matn2, and Igf2) were able to stimulate CM proliferation individually. Of notice, all these proteins have been shown to promote the proliferation of other cell types, in particular cancer cells, and a few of them have been implicated in other regenerative processes. Cystatin F (Cst7) is a member of the cystatin super-family and is predominantly expressed by the cells of the hematopoietic lineage, as well as by metastatic cancerous cells, although it is not yet clear whether it favors seeding, survival or proliferation of tumor cells at secondary sites (mainly liver and spleen)[22,23]. TNF superfamily member 11 (Tnfsf11), or RANKL, is a member of the tumor necrosis factor (TNF) cytokine family, which is a ligand for osteoprotegerin and functions as a key factor for osteoclast differentiation and activation. It is also a key paracrine effector of progesterone signaling, which importantly contribute to mammary tumorigenesis[24]. Interleukin-33 (Il33)—a member of the IL-1 family—was originally described as an inducer of type 2 immune responses, activating T helper 2 cells and mast cells. More recently, evidence has accumulated that this factor is more pleiotropic in nature, bridging innate and adaptive immunity in the regulation of tissue homeostasis, injury and repair[25]. The interaction between Il33 and its membrane receptor ST2L is enhanced in response to myocardial stress and exerts cardioprotective actions in the myocardium by reducing fibrosis and hypertrophy. In accordance, the circulating, soluble isoform of the receptor (sST2), by sequestering Il33, abrogates these favorable activities and is currently considered among the potential biomarkers for heart failure[26]. Although a direct activity of Il33 in enhancing cardiomyocyte proliferation has never been reported so far, its capacity to foster the growth and metastatic invasion of at least some cancer types, including non-small-cell lung cancer (NSCLC)[27], epithelial ovarian cancer[28], breast cancer[29], and glioma[30]. Fibrinogen-like 2 (Fgl2), also known as fibroleukin, is an additional multifunctional protein, involved in a variety of physiological and pathological processes, including viral infections, pregnancy failure, autoimmune disorders, allograft rejections, and tumor growth[31]. While inhibiting T cell proliferation in various contexts[32], it contributes to hepatocellular carcinoma (HCC) tumor growth and angiogenesis[33] in a thrombin-dependent manner, thus indicating that its activity in vivo strictly depends on target cells and local microenvironment. Matrilin2 (Matn2) is the largest member of the matrilin family of multidomain adapter proteins, which interact with other extracellular matrix proteins to form filamentous networks by connecting proteoglycans and collagen fibrils[34]. Matn2 forms

oligomers, which are deposited in connective tissues of almost any organ and is required for both peripheral nerve[35] and muscle regeneration[36]. Of interest, in the muscle Matn2 is deposited around proliferating, differentiating, and fusing myoblasts in culture and during muscle regeneration in vivo, where it plays a key role in regulating the cascade that initiated terminal myogenic differentiation[36]. Whether a similar mechanism may occur during mammalian heart regeneration is an interesting possibility, which deserves further investigation. Insulin-like growth factor 2 (IGF2) is a protein hormone known to regulate cell proliferation, growth, migration, differentiation, and survival. Beside its growth-promoting and beneficial functions during embryonic development and placental growth, a role for IGF2 has been clearly established in both cancer and cardiovascular diseases[37]. Even though data are not always consistent, IGF2 has been shown to act in an autocrine or paracrine fashion to promote the growth of a variety of neoplasia, including brain tumors, mammary carcinoma, pancreatic carcinoma, and ovarian carcinoma[37]. It also acts as a pivotal promoter of growth of atherosclerotic lesions in mice and its local overexpression induces smooth muscle proliferation and the appearance of aortic focal intimal masses[38].

Thus, any of these six factors appears able to exert a pro-proliferative effect on various cell types. Our data do not prove the necessity of these factors in mediating the capacity of Tregs to stimulate CM proliferation. However, the administration of AAV vectors expressing the pool of Treg-specific pro-proliferative cytokines recapitulated both the beneficial effect on heart function following myocardial infarction and incorporation of EdU by CMs, thus pointing toward these six factors as relevant players in the beneficial effect of Tregs on the damaged heart.

## Methods

**Isolation and culture of neonatal and adult cardiomyocytes.** Neonatal cardiomyocytes were isolated and cultured from p0-p1 old pups by enzymatic digestion[15]. Briefly, ventricles from neonatal rats were separated from the atria and enzymatically dissociated and digested in CBFHH basal buffer (calcium and bicarbonate-free Hanks buffer with Hepes) containing Trypsin (1.8 mg/ml, Gibco), DNAse II (2 mg/l, Sigma Aldrich), and Gentamicin (1%, Sigma Aldrich) for 3 h at 37 °C under gentle agitation. Trypsin activity was stopped by adding 8 ml of fetal bovine serum (FBS) to the collected cells. The solution was centrifuged for 10 min at $300 \times g$, and the re-suspended pellet filtered through a 40-μm cell strainer (BD Biosciences) before seeding. Cardiomyocytes were enriched (>90% purity) over non-myocytes by a 2-h pre-plating step on non-primary 100-mm dishes in complete medium (DMEM 4.5 g/L of D-Glucose, Gibco), supplemented with 5% FBS (Gibco), 2 mg/ml vitamin B12 (Sigma Aldrich), 100 U/ml penicillin, and 100 U/ml streptomycin (Sigma Aldrich). Myocytes, either in solution or lightly attached, were separated from the stromal cells by gentle mechanical disaggregation, counted, and subsequently plated on primary plates at a density of $1.5 \times 10^5$ cells/ml. After 24 h, the medium was changed and cells were subjected to the different treatments. Adult cardiomyocytes were isolated and cultured according to a Langendorff-free procedure[39]. All experimental conditions were tested in triplicate on at least three independent cell cultures.

Cell proliferation was evaluated by administration of EdU (10 mM, Life Technologies) for 20 h when testing proliferation in CMs only (Figs. 1 and 6) and for 10 h when also testing proliferation in fibroblasts (Fig. 2), followed by washing, fixation and immunofluorescence staining.

**Isolation and culture of Tregs.** Tregs were isolated from mesenteric, axillary, inguinal, and cervical lymph nodes using the $CD4^+CD25^+$ Regulatory T Cell Isolation Kit (Miltenyi Biotec), using a two-step procedure. First, non-$CD4^+$ cells were magnetically labeled with a cocktail of biotin-conjugated antibodies against CD8, CD11, CD45R, CD49b, Ter-119, and Anti-Biotin MicroBeads. The labeled cells were subsequently depleted using a MACS LS column (Miltenyi Biotec). In the second step, the flow-through fraction of pre-enriched $CD4^+$ T cells was labeled with CD25 MicroBeads for subsequent positive selection of $CD4^+CD25^+$ Tregs. Isolated cells were counted and seeded into 96-well round bottom plates, previously activated overnight at 4 °C with an anti-mouse CD3 antibody in PBS (1:1000, Invitrogen) and cultured in RPMI-1640 supplemented with 100 U/ml penicillin, 100 μg/ml streptomycin, 10% FBS, and recombinant murine IL2 (50 ng/ml). The supernatants produced by Treg cells were recovered at 24–48–72 h and stored at −80 °C.

**Animal studies.** Animal care and treatment were conducted in conformity with institutional guidelines in compliance with national and international laws and policies (European and Economic Council Directive 86/609, OJL 358, December 12, 1987), upon approval by the ICGEB Institutional Animal Welfare Board and by the Italian Ministry of Health. CD1, athymic nude Foxn1, Fox Chase SCID, and Fox Chase SCID BEIGE mice were purchased from Harlan Laboratories. EGFP (C57BL/6-Tg(CAG-EGFP)1Osb/J) mice were purchased from The Jackson Laboratory. DEREG[40,41] and Colα1(I)-EGFP mice[42] were bred in house. A block randomization scheme was used to assign animals to groups on a rolling admissions basis to obtain adequate samples for each time point and each experiment.

To assess CM proliferation in vivo animals received EdU (350 μg per animal, intraperitoneally) every 2 days for the indicated time periods.

Blood was drawn from the jugular vein of anesthetized animals and allowed to clot at 37 °C for 2 h, followed by centrifugation at $200 \times g$ for 10 min at 4 °C. Sera were immediately transferred into a new tube with the addition of 100 U/ml penicillin and 100 g/ml streptomycin to avoid bacterial contamination and stored at −20 °C until use.

**Myocardial infarction.** Mice (2-month-old female animals, $n = 8$ per group) were anesthetized by intraperitoneal injection of ketamine–xylazine and laid down in a supine position on a dedicated pad at 37 °C, fixed to the plate and intubated. To reach the anterior wall of the heart, a xifo-axillar incision was made, exposing the underlying muscles. The pectoralis major muscle was lifted up and fixed with a retractor, while the underlying pectoralis minor was cut to expose the ribs. The fifth intercostal space was pierced and enlarged with a retractor opening the thorax. The pericardium was stripped exposing the heart anterior wall. The left anterior descending coronary artery was identified and ligated 1 mm below the left atrium auricula. Effective ligation of the coronary artery was confirmed by whitening of the heart anterior wall. Ligation was either left in place permanently or removed after 40 min to allow reperfusion after ischemia. All treatments (Tregs and AAV-pools) were injected into the left ventricle anterior wall, at the border region of the infarct, using a 0.3 ml insulin syringe with a 30-gauge needle. All anatomical structures were visualized with a stereomicroscope (Leica). Intercostal spaces, muscles and skin were sutured and mice were extubated to re-establish normal breathing. Mice were then laid in a prone position and kept on the warmed pad until awakening and later transferred to a new cage.

To evaluate heart function and size, transthoracic two-dimensional echocardiography was performed in mice anaesthetized with isoflurane keeping the heart rate over 450 bpm, using a Vevo 2100 Ultrasound (Visual Sonics) equipped with a MS550D 22–50 MHz linear array solid-state transducer. B-mode multiplanar tracings in parasternal short and long axis views (modified Simpson's method) were used to measure left ventricular anterior and posterior wall thickness, septum thickness and left ventricular internal diameter at end-systole and end-diastole, which were used to calculate left ventricular fractional shortening and ejection fraction.

**Treg depletion.** For depletion experiments, purified anti-CD25 monoclonal antibodies (clone PC61, ATCC, 1 mg per mouse) were injected intraperitoneally into CD1 mice. Alternatively, diphtheria toxin (Calbiochem, 500 ng per mouse) was administered intraperitoneally once a week in DEREG mice. Efficacy of depletion was assessed in mesenteric, axillary, inguinal, and cervical lymph nodes by flow cytometry.

**Flow cytometry and cell sorting.** For Treg staining and cell sorting, cells were analyzed using either a BD FACSCalibur or a FACS Aria II (BD), incubating cells on ice for 30 min in PBS and 2% BSA with PE-conjugated anti-CD25 antibodies (clone 7D4, Miltenyi Biotec #130102788) or PerCP-conjugated anti-CD4 (clone RM4-5, BD Pharmingen #553052) antibodies (1:100 dilution for both). Analysis of flow cytometry data was performed using FlowJo (version 4.5.4; Tree Star). Isotype controls were performed with corresponding rat Ig.

**Immunofluorescence and histology.** Immunofluorescence was performed on both tissue sections and plated cells. After euthanasia, mouse hearts were excised, briefly washed in PBS and either fixed in 10% formalin at room temperature or snap frozen in isopentane/liquid nitrogen. Formalin-fixed tissues were embedded in paraffin and cut into 4 μm tissue sections, de-waxed in xylene for 30 min and rehydrated with alcohols at decreasing concentration (100, 90, 70, 50%) at room temperature. Antigen retrieval was performed on sections by boiling 20 min in 0.1 M sodium citrate buffer solution at pH 6.0 and letting cool down at room temperature for 3 h. Sections were rinsed three times in water, permeabilized 30 min in 0.5% Triton X-100 PBS, and then blocked for 1 h in 20% horse serum PBS. Cells were grown on gelatin–fibronectin-coated cover slips and fixed in 4% paraformaldehyde for 15 min at room temperature.

All samples were washed three times in PBS, permeabilized in 0.5% Triton 100-X for 30 min, and blocked in 10% goat serum for 1 h. Sections were stained overnights at 4 °C with the following antibodies: anti-PCM1 (Sigma Aldrich #HPA023374), anti-sarcomeric α-actinin (Abcam #9465), anti-aurora B (Abcam #2254), anti-phospho-histoneH3-Ser10 (Millipore #06-570), anti-Ki-67 (Monsan #PSX1028), anti-EGFP (Abcam #6658), anti-CD45 (BD Pharmingen # 550539), all

1:100 in 5% goat serum. After two washing steps of 5 min in 0.5% Triton X-100 at room temperature, sections were incubated 1 h in 1:200 secondary antibody conjugated to Alexa Fluor-488 or Alexa Fluor-594 (Life Technologies) in 10% goat serum for 45 min at room temperature. EdU was detected using the Click-iT EdU Alexa Fluor® 594 Imaging Kit (Invitrogen). Nuclei were stained with a 0.1 mg/ml l of 4′, 6-diamidino-2-phenylindole dihydrochloride (DAPI) solution (Sigma).

Azan-Mallory's trichrome staining (Bio Optica) was performed according to standard procedures and analyzed for morphology and extension of fibrosis. Infarct size was calculated as the percentage of the total left ventricular area showing fibrosis.

**TUNEL assay.** Myocardial apoptosis was examined using the In Situ Cell Death Detection Kit, TMR red (Roche). Briefly, 5 μm-thick heart sections were incubated with the labeling mixture supplied by the manufacturer at 37 °C for 1 h. Cell nuclei were counterstained with Hoechst 33342 (Invitrogen). Three slides from each block were evaluated for percentage of apoptotic cells and four fields on each slide were examined at the border areas. The number of apoptotic CMs is presented as a percentage of total CMs.

**RNA isolation and quantitative real-time PCR.** Total RNA was extracted using TRIzol (ThermoFisher) and reverse transcribed using hexameric random primers. The amplifications were performed on a BioRad Real-time thermal cycler CFX96 machine, using the following TaqMan probes (Applied Biosystems): mTNFα Mm00443258_m1, mMMP9 Mm00442991_m1, mTGFβ1 Mm01178820_m1, mFoxp3 Mm00475162_m1, mIL10 Mm00439614_m1, mIL6 Mm00446191_m1, mIFNγ Mm01168134_m1, and mIl12p40 Mm01288989_m1. The murine house-keeping GAPDH gene (mGAPDH Mm99999915_g1) was used to normalize the results.

**CHO cell transfection and supernatant collection.** Chinese Hamster Ovary (CHO) cells were cultured in F12 medium (F12 Nutrient Mixture, Thermo Fisher Scientific) supplemented with 10% FBS (GIBCO), L-glutamine 2 mM, 100 U/ml penicillin (Sigma Aldrich) and 100 μg/ml streptomycin (Sigma Aldrich).

Transfection was carried out in 24-well plates using Lipofectamine 2000 (Invitrogen, Thermo Fisher Scientific). Cells were transfected with 0.4 μg plasmid DNA in a 1:2 μg DNA: μl Lipofectamine ratio. After 24 h, cells were washed to remove FBS and kept in serum-free medium for additional 24 h. Supernatants produced by the transfected cells were collected, centrifuged to remove any cell and stored at −80 °C. The list of plasmids encoding for the soluble factors expressed by Tregs is reported in Table 1.

**AAV production.** Recombinant AAV vectors were prepared in the AAV Vector Unit at ICGEB Trieste, (www.icgeb.org/avu-core-facility)[43]. Briefly, AAV vectors were generated in HEK293T cells, using a triple-plasmid co-transfection for packaging. Viral stocks were obtained by CsCl₂ gradient centrifugation. Titration of AAV viral particles was performed by real-time PCR quantification of the number of viral genomes. The viral preparations had titers between $2 \times 10^{12}$ and $3 \times 10^{13}$ vg/ml.

**Statistics.** Data are expressed as the mean of biological replicates, and error given as standard error of the mean. Statistical analysis was performed using the SPSS software considering a $P$-value of less than 0.05 as statistically significant. For histological analysis and gene expression data at a single time point, the statistical significance of the differences between groups was determined using the unpaired $t$ test. For morphological and functional scores among the time points within each group and among the groups within each time point we used two-way anova for repeated measurements, followed by Tukey's pairwise post-hoc test or Bonferroni/Dunn's post-hoc test. An $F$-test was used to compare variance. For in vivo experiments, a statistical design of the sample size was performed using the software http://homepage.stat.uiowa.edu/~rlenth/Power/, setting a variation coefficient ($s$) of 30%, a minimal relative effect ($\delta$) of 30%, alpha of 5%, and a power of 80% ($p$).

**Data availability.** All data generated or analyzed during this study are included in this article and its supplementary information files. The transcriptome of CD4⁺/CD25⁺ Tregs with that of CD4⁺/CD25⁻ lymphocytes was derived from the GEO database (https://www.ncbi.nlm.nih.gov/geo/query/acc.cgi?acc=GSE4571). All relevant data are available from the authors upon reasonable request.

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

## Acknowledgements

This work was supported by grant 14CVD04 from the Leducq Foundation Transatlantic Network of Excellence and grant RF-2011-02348164 "Cardiorigen" from the Italian Ministry of Health to M.G., and by grant AIRC IG 2016 19032 to S.Z. D.K. was supported by the LabEx Transimmunom (ANR-11-IDEX-0004-02) and ERC Advanced Grant TRiPoD (322856). We are grateful to T. Sparwasser for having provided DEREG mice.

## Author contribution

S.Z. designed the experiments, performed in vivo studies and prepared the manuscript; V.M., S.M. and Al.C. performed cell culture studies; An.C. performed immuno-fluorescence and image acquisition; A.N. performed the experiments with the immune-compromised animals; M.R. performed Trichrome staining, M.A., S.V., G.C. and E.D. contributed to in vivo experiments and echocardiography, C.P. performed flow cyto-metry; L.Z. produced the AAV vectors; M.I.G. produced the PC61 antibodies; C.L. provided the Collα1(I)-EGFP mice and contributed to the analysis of the results, G.S., D. K. and M.G. critically reviewed the design and results of the study.

## Additional information

**Competing interests:** The authors declare no competing interests.

