## [Peer Review File · Nature Communications]

Reviewers' comments:

Reviewer #1 (Remarks to the Author):

In this manuscript, authors investigate a potential role of regulatory T cells in promoting cardiomyocyte proliferation in dams during pregnancy, and in the context of cardiac ischemia.

Previous studies have demonstrated an increase in maternal heart size during pregnancy, attributed to hypertrophic remodeling. Previous studies have also addressed the role of regulatory T cells during ischemic cardiac injury. The potential novelty of this study is a potential role of regulatory T cells in stimulation of cardiomyocyte proliferation. Unfortunately, however, quality of data presented is not sufficient to make the case that regulatory T cells promote cardiomyocyte proliferation. In particular, as neonatal cardiomyocyte cultures are not pure myocytes, containing cardiac stromal populations that are small, and as cardiac tissue in the same manner has a complexity of smaller stromal cells that more readily proliferate than cardiomyocytes, it is critical to use a nuclear marker of cardiomyocytes to show coincidence of a proliferative marker with myocyte nuclei (see Soonpaa and Field for review and discussion). EdU incorporation may signify DNA synthesis, which can occur during hypertrophy, or DNA damage repair, rather than proliferation. In addition, many of the presumably "representative" images shown do not seem to support accompanying quantitative data, both for in vitro and in vivo data. Additionally, to demonstrate data more clearly, lower magnification images of tissues should also be shown in addition to high magnification images, to allow the reviewer/reader to assess how representative the higher magnification images are. Adequate experimental controls are sometimes lacking.

Specific Comments

1. Nuclear cardiomyocyte marker should be used to show coincidence of proliferative marker and myocytes throughout the manuscript
2. Fig. 1E, lower magnification images of tissue should be shown, as well as higher magnification. Images should be taken from comparable regions of the heart, and distinct compartments. How do right ventricle, left ventricle, septum, atria, compare?
3. Fig. 1 F, images shown for Ki67 and H3P do not show much if any labeling of any cardiomyocyte nuclei? There is one potential myocyte nucleus shown in high magnification for H3P, but why are high magnification images only shown for TregSN, not for controls?

4. Fig. 2. DNA synthesis can occur during hypertrophy. If cardiomyocytes are incorporating more EdU, this may be owing to increased DNA synthesis, rather than proliferation. Quantification of Aurora B kinase staining is not presented. Data on control female hearts is not presented.
5. Fig. 3. Data assessing fibrosis are very weak. Only GFP staining, and only high magnification image shown. Trichrome staining and quantification of fibrotic area should complement this assessment.
6. Fig. 4. Are T regs in the vicinity of EdU incorporating cardiomyocytes? Co-immunostainings should be performed. EdU incorporation should be quantified. EdU staining cells shown in G may or may not be cardiomyocytes. Control heart images should also be shown for comparison.
7. Fig. 4E. Why does the control have such a depressed ejection fraction at 1 week, this is different from other controls shown for similar physiological procedure in preceding figure? Suggests variability of the procedure. Thus, the n of 8 that is cited in Materials and Methods may be insufficient?
8. Fig. 5F. AAV9 alone controls are lacking?
9. In several panels, or sometimes in entire figures (such as Figs. S2 and S7), alpha-actinin stainings are overexposed making it look like the myocardial wall is exclusively composed of myocytes. In Fig. S7D, areas that are clearly fibrotic also seem to have alpha-actinin staining, showing this staining is not a good strategy for identifying myocyte nuclei. These observations reinforce the absolute need of using a nuclear myocyte marker (such as PCM1) to perform studies quantifying myocyte proliferation.

Reviewer #2 (Remarks to the Author):

Zacchigna et al have studied the role of maternal Tregs in induction of cardiomyocyte proliferation during development, on the maternal heart during pregnancy and after myocardial infarction. Using a variety of mouse models, they demonstrate that Treg conditioned media promotes neonatal rat cardiomyocyte proliferation and that Treg depletion inhibits fetal and maternal cardiomyocyte proliferation. In the adult heart, Tregs were shown to accumulate rapidly post-infarction, to persist for at least a week, and to synthesize a variety of cytokines. Treg depletion worsened cardiac repair and function, whereas transplantation of Tregs enhanced cardiac repair and function. Finally, AAV overexpression of Treg cytokines induce cardiomyocyte proliferation post infarction and enhance ventricular function.

General Comments

This is an interesting paper that has several surprising findings about Tregs. The authors are commended for the multidisciplinary manner in which the study was conducted. Although many questions are left unanswered, this study advances our understanding of maternal-fetal cardiac biology and adult myocardial infarction. I have some suggestions for improvement.

1. The manuscript implies that the 6 factor cocktail explains the beneficial effects of Tregs on the infarcted heart. The studies point to sufficiency, but they do not address the necessity of these factors. Please include this in the discussion.
2. It would be interesting to know what the effect of Treg depletion was on maternal cardiomyocyte proliferation during pregnancy. Are these data available?
3. The basal rates of EdU incorporation in the heart seem quite high, ~0.2%. Please reconcile this with the literature showing significantly lower rates.
4. Larger infarcts in the Treg depleted hearts could result from increased cell death at the border zones with infarct extension. Please check for differences in cell death.
5. A few pieces of information would help non-immunologists understand the paper.
 - a. In pregnancy, is it simply a difference in Treg numbers, or are there important qualitative differences?
 - b. When do mice begin producing Treg cells? Do maternal Tregs cross the placenta? Will the depleting antibody cross the placenta? Will Treg-derived cytokines cross the placenta?

Specific Points

1. Abstract. Please give some numbers to help understand the magnitude of the Treg impact in various contexts.
2. Please present both individual and group data for the in vivo studies, so that readers get a sense of the data distribution.
3. Are the pregnancies inbred or intercross matings? This seems to be important in the Treg depletion studies.
4. For the Treg conditioned medium experiments, do the controls include IL-2 that is used as a Treg stimulating cytokine?

5. Pregnancy induces more of a length change of cardiomyocytes than a diameter change, since it is a volume overload. Please mention that in the section on cross sectional diameter analysis.

Reviewer #3 (Remarks to the Author):

The manuscript by Zacchigna et al. describes a putative role of T regulatory lymphocytes in the regulation of cardiomyocyte proliferation, and links such activity to the observed capability of cardiomyocytes to proliferate during development and shortly after birth (in mice). Overall this is an interesting question as it is currently not well understood how the regenerative phenotype in newborn mice comes about, and what factors are responsible for the incapability of adult cardiomyocytes to proliferate. The manuscript follows an original idea but is flawed by methodological concerns that preclude convincing proof for the authors' hypothesis. The positive effect of Tregs on post-MI recovery is not novel. The assumption that permanent occlusion causes substantially less inflammation than ischemia reperfusion ("Indeed, it is commonly believed that inflammation plays a major role in cardiac damage after ischemia reperfusion while it is less relevant in permanent ischemia.

") is wrong and shows that the authors should catch up on the relevant literature on post-MI inflammation.

The uncritical use of the DREG mouse (and antibodies) to deplete Tregs is a flawed experiment. Such a depletion leads to a systems wide autoimmune disease and actually kills the mouse after 2-3 weeks because of general inflammation. Thus, the worse post-MI outcome may be unrelated to any influence of Tregs on cardiomyocyte proliferation. The tissue analysis also appears subpar and does not match previous reports on the frequency and time course of Tregs after MI (e.g. suppl Fig 3 seems to implicate that Tregs are recruited very early while previous publications using state of the art FACS showed different results). The in vitro exposure of cardiomyocytes to serum of pregnant mice is interesting, however Tregs do not cross the placenta.

The association of Treg peak in circulation and cardiomyocyte proliferation does not prove causality.

Fig. 2E: please improve the unclear group assignment.

Please add page numbers to your manuscript.

Below is a point-by-point response to the issues raised by the reviewers. The reviewer's text is in black, our response in blue.

Reviewer #1 (Remarks to the Author):

In this manuscript, authors investigate a potential role of regulatory T cells in promoting cardiomyocyte proliferation in dams during pregnancy, and in the context of cardiac ischemia.

Previous studies have demonstrated an increase in maternal heart size during pregnancy, attributed to hypertrophic remodeling. Previous studies have also addressed the role of regulatory T cells during ischemic cardiac injury. The potential novelty of this study is a potential role of regulatory T cells in stimulation of cardiomyocyte proliferation. Unfortunately, however, quality of data presented is not sufficient to make the case that regulatory T cells promote cardiomyocyte proliferation. In particular, as neonatal cardiomyocyte cultures are not pure myocytes, containing cardiac stromal populations that are small, and as cardiac tissue in the same manner has a complexity of smaller stromal cells that more readily proliferate than cardiomyocytes, it is critical to use a nuclear marker of cardiomyocytes to show coincidence of a proliferative marker with myocyte nuclei (see Soonpaa and Field for review and discussion). EdU incorporation may signify DNA synthesis, which can occur during hypertrophy, or DNA damage repair, rather than proliferation. In addition, many of the presumably "representative" images shown do not seem to support accompanying quantitative data, both for in vitro and in vivo data. Additionally, to demonstrate data more clearly, lower magnification images of tissues should also be shown in addition to high magnification images, to allow the reviewer/reader to assess how representative the higher magnification images are. Adequate experimental controls are sometimes lacking.

Specific Comments

1. Nuclear cardiomyocyte marker should be used to show coincidence of proliferative marker and myocytes throughout the manuscript

We thank the Reviewer for this instructive suggestion. Accordingly, we have introduced anti-PCM1 staining in most of the figures to label cardiomyocyte nuclei. However, as we show in Supplementary Figure 2, even this marker is not 100% specific, as various neonatal fibroblasts also score positive, whereas several adult cardiomyocyte nuclei score negative. Therefore, we believe that the two stainings (\$\alpha\$ -actinin and PCM1) could provide complementary information and add reliability in the identification of cardiomyocytes in culture and in vivo. Therefore, we have not replaced \$\alpha\$ -actinin staining with PCM1, but rather added a series of PCM1 stainings and quantifications throughout the manuscript.

2. Fig. 1E, lower magnification images of tissue should be shown, as well as higher magnification. Images should be taken from comparable regions of the heart, and distinct compartments. How do right ventricle, left ventricle, septum, atria, compare?

As suggested by the Reviewer, we have assessed proliferation rate in the distinct compartments of the heart in both normal and Treg-depleted fetal hearts, using both \$\alpha\$ -actinin and PCM1 as CM-specific markers. These data are presented in the new Figure 1E, F. We have not included low magnification pictures, as proliferating

nuclei could not be properly appreciated.

3. Fig. 1 F, images shown for Ki67 and H3P do not show much if any labeling of any cardiomyocyte nuclei? There is one potential myocyte nucleus shown in high magnification for H3P, but why are high magnification images only shown for TregSN, not for controls?

We apologize for the low quality of previous pictures. We have now replaced previous Figure 1F with Figure 2, in which we added high magnification images for all conditions to comply with the Reviewer's recommendation.

4. Fig. 2. DNA synthesis can occur during hypertrophy. If cardiomyocytes are incorporating more EdU, this may be owing to increased DNA synthesis, rather than proliferation. Quantification of Aurora B kinase staining is not presented. Data on control female hearts is not presented.

As suggested by the Reviewer, we added control female hearts (not pregnant) in the new Figure 3, panels E, G and H, and also included quantification of Aurora B+ cardiomyocytes.

5. Fig. 3. Data assessing fibrosis are very weak. Only GFP staining, and only high magnification image shown. Trichrome staining and quantification of fibrotic area should complement this assessment.

Following the Reviewer's recommendations, we now provide more reliable assessment of fibrosis by showing a few representative Masson Trichrome stainings for both control and Treg-depleted animals (New Figure 4, panel D) and quantification of the extension of the fibrotic area (panel E).

6. Fig. 4. Are T regs in the vicinity of EdU incorporating cardiomyocytes? Co-immunostainings should be performed. EdU incorporation should be quantified. EdU staining cells shown in G may or may not be cardiomyocytes. Control heart images should also be shown for comparison.

It was not easy to detect EdU+ cardiomyocytes in the vicinity of the injected Tregs in our experiments, as we analyzed Treg engraftment and CM proliferation at different times. Indeed, we have shown clear presence and survival of the injected cells at day 3 (new Figure 5A), but we do not expect long-term engraftment and indeed very few cells were still detectable at 2 weeks, when we have assessed cardiomyocyte proliferation and cardiac function. However, we could find a few cases of EGFP+ Tregs in close proximity of EdU+ cardiomyocytes and we have now added a representative example in Figure 5H. In addition, again following the recommendations of this Reviewer, we have identified proliferating cardiomyocytes by EdU and PCM1 co-staining. These results are shown in Figure 5G.

7. Fig. 4E. Why does the control have such a depressed ejection fraction at 1 week, this is different from other controls shown for similar physiological procedure in preceding figure? Suggests variability of the procedure. Thus, the n of 8 that is cited in Materials and Methods may be insufficient?

The Reviewer refers to experiments performed in current Figures 4 and 6 on CD1 mice and in current Figure 5 on C57/BL6 mice. It is not unusual that different strains and even different litters of the same strain respond differently to LAD ligation. This is the reason why we always include, for each experimental session, both controls and treated animals from the same litter, blinding their assignment to the different experimental groups.

8. Fig. 5F. AAV9 alone controls are lacking?

As suggested by the Reviewer, we have added images from control mice in the new Figure 6F.

9. In several panels, or sometimes in entire figures (such as Figs. S2 and S7), alpha-actinin stainings are overexposed making it look like the myocardial wall is exclusively composed of myocytes. In Fig. S7D, areas that are clearly fibrotic also seem to have alpha-actinin staining, showing this staining is not a good strategy for identifying myocyte nuclei. These observations reinforce the absolute need of using a nuclear myocyte marker (such as PCM1) to perform studies quantifying myocyte proliferation.

Embracing the Reviewer's suggestions, we removed the old Supplementary Figure S2 and added new images to the main Figure 3G. Throughout the manuscript we have included PCM1 staining to label CMs. Since previous Supplementary Figure 7 (now Supplementary Figure 9) shows lack of CM proliferation, we did not feel the need to add this analysis in this Supplementary Figure.

Reviewer #2 (Remarks to the Author):

Zacchigna et al have studied the role of maternal Tregs in induction of cardiomyocyte proliferation during development, on the maternal heart during pregnancy and after myocardial infarction. Using a variety of mouse models, they demonstrate that Treg conditioned media promotes neonatal rat cardiomyocyte proliferation and that Treg depletion inhibits fetal and maternal cardiomyocyte proliferation. In the adult heart, Tregs were shown to accumulate rapidly post-infarction, to persist for at least a week, and to synthesize a variety of cytokines. Treg depletion worsened cardiac repair and function, whereas transplantation of Tregs enhanced cardiac repair and function. Finally, AAV overexpression of Treg cytokines induce cardiomyocyte proliferation post infarction and enhance ventricular function.

General Comments

This is an interesting paper that has several surprising findings about Tregs. The authors are commended for the multidisciplinary manner in which the study was conducted. Although many questions are left unanswered, this study advances our understanding of maternal-fetal cardiac biology and adult myocardial infarction. I have some suggestions for improvement.

We thank the Reviewer for his/her appreciation of our work and multidisciplinary approach.

1. The manuscript implies that the 6 factor cocktail explains the beneficial effects of Tregs on the infarcted heart. The studies point to sufficiency, but they do not address the necessity of these factors. Please include this in the discussion.

As suggested by the Reviewer, we have now included this concept in the discussion.

2. It would be interesting to know what the effect of Treg depletion was on maternal cardiomyocyte proliferation during pregnancy. Are these data available?

In the revised version of the manuscript, we have introduced a new Supplementary Figure 3, showing that Treg depletion results in reduced CM proliferation in the mother's heart.

3. The basal rates of EdU incorporation in the heart seem quite high, ~0.2%. Please

reconcile this with the literature showing significantly lower rates.

We are aware the rate of EdU incorporation is variably reported in the literature, possibly depending on the dose and frequency of administration. It is recognized that EdU incorporation increases significantly after damage, such as a myocardial infarction, and indeed our basal rate always refer to infarcted hearts. Our data are consistent with previously published papers by our (Eulalio et al, Nature 2012;492(7429):376-81; Lesizza et al. Circ Res. 2017;120:1298-1304) and other groups (Leach, 2017 Nature 550, 260–264).

4. Larger infarcts in the Treg depleted hearts could result from increased cell death at the border zones with infarct extension. Please check for differences in cell death. According to the Reviewer's suggestions, we have performed a TUNEL assay showing no significant differences in cell deaths in control and Treg-depleted hearts. This information is now included in Supplementary Figure 6.

5. A few pieces of information would help non-immunologists understand the paper.

a. In pregnancy, is it simply a difference in Treg numbers, or are there important qualitative differences?

b. When do mice begin producing Treg cells? Do maternal Tregs cross the placenta? Will the depleting antibody cross the placenta? Will Treg-derived cytokines cross the placenta?

We thank the Reviewer for having addressed these interesting questions.

a) There is a consensus that an overall expansion of the maternal Treg compartment occurs during pregnancy. In mice, there is an increase in Tregs within the uterus during the oestrus cycle that prepares the uterus for a potential embryo implantation. If the implantation occurs, the expansion continues, first involving only thymic derived Tregs, and later involving peripherally induced Tregs starting at E14. Nothing is known yet about changes in function (specific work on this matter is ongoing in the laboratory of Prof. Klatzmann, who is a co-author of the present manuscript), but no published data are available yet). To the best of our knowledge, in humans Treg studies have only been performed with cells from peripheral blood. No differences in their functionality have been described.

b) T cell development in the thymus takes approximately 3 weeks, the same time as pregnancy. At birth, the first mature T cells start seed in the newborns. For unknown reasons, there is a delay in the seeding of Tregs compared to conventional T cells. During the first 3 postnatal days, only T effector cells are released, thereby explaining why thymectomy during this period leads to autoimmunity due to the absence of Tregs.

There is a well-documented microchimerism between mother and child. On the one hand, CD3+ or CD34+ cells carrying the Y chromosome have been found in almost all tissues of pregnant women who died during pregnancy of a male fetus (Rijnink et al; MHR 2015). On the other hand, some maternal cells populate fetal lymph nodes and have been shown to induce tolerance to maternal alloantigens. As this tolerance is decreased if maternal Tregs are depleted, it has been assumed that maternal Tregs somehow populate the fetus (J. Mold, Science, 2008). However, to our knowledge, there is no direct proof that maternal Tregs can either cross or populate the placenta. In contrast, and relevant to this manuscript, both Treg-depleting antibodies and Treg-derived cytokines cross the placenta. We have now added most of this information in the Discussion.

Specific Points

1. Abstract. Please give some numbers to help understand the magnitude of the Treg impact in various contexts.

As recommended some quantifications are provided in the abstract.

2. Please present both individual and group data for the in vivo studies, so that readers get a sense of the data distribution.

As requested we have changed the format of the graphical representation of most in vivo data, showing both individual and group data.

3. Are the pregnancies inbred or intercross matings? This seems to be important in the Treg depletion studies.

Pregnancies in this study came from intercross mating. We have intentionally included both the DREG model, which is an inbred strain on a C57BL/6 background and the anti-CD25 model in CD1 outbred mice. As expected, the most interesting data came from CD1 outbred mice.

4. For the Treg conditioned medium experiments, do the controls include IL-2 that is used as a Treg stimulating cytokine?

We apologize for the lack of clarity and we have added this information in the new legend to Figure 2.

5. Pregnancy induces more of a length change of cardiomyocytes than a diameter change, since it is a volume overload. Please mention that in the section on cross sectional diameter analysis.

As suggested, we have mentioned this concept in the indicated section.

Reviewer #3 (Remarks to the Author):

The manuscript by Zacchigna et al. describes a putative role of T regulatory lymphocytes in the regulation of cardiomyocyte proliferation, and links such activity to the observed capability of cardiomyocytes to proliferate during development and shortly after birth (in mice). Overall this is an interesting question as it is currently not well understood how the regenerative phenotype in newborn mice comes about, and what factors are responsible for the incapability of adult cardiomyocytes to proliferate. The manuscript follows an original idea but is flawed by methodological concerns that preclude convincing proof for the authors' hypothesis. The positive effect of Tregs on post-MI recovery is not novel. The assumption that permanent occlusion causes substantially less inflammation than ischemia reperfusion ("Indeed, it is commonly believed that inflammation plays a major role in cardiac damage after ischemia reperfusion while it is less relevant in permanent ischemia.") is wrong and shows that the authors should catch up on the relevant literature on post-MI inflammation.

While recognizing that our sentence could appear naïf to an expert reader, we do not agree on our lack of update on the relevant literature on post-MI. Indeed, we co-authored a position paper of the ESC Working Group on Myocardial Function specifically reviewing the role of the immune activation following myocardial infarction (this manuscript has been accepted for publication on EurJHeartFailure a few weeks ago). As such, we are aware that multiple inflammatory (i.e. TLRs, NF-κB,

TNF α) pathways are activated after permanent coronary artery ligation. On the other hand, there is solid experimental evidence that both T- and B-lymphocytes, but most importantly CD4+ T-cells, enhance ischemia–reperfusion injury, although the latter are also required for myocardial healing (Hofmann and Frantz, *Circ Res* 2015, 116(2)).

We believe that the Reviewer refers to an elegant paper showing the temporal dynamics of cardiac immune cell accumulation following acute myocardial infarction by flow cytometry. This paper showed that a timely reperfusion reduced the total number of leukocytes accumulated in the post-MI period, shifting the peak of innate immune response towards earlier (Yan et al., *JMCC* 2013, 62:24-35) and resulting in reduced infarct size after ischemia-reperfusion compared to permanent ligation. However, this work essentially characterized the early phase after myocardial infarction (up to 7 days for flow cytometry and 28 days for histology). In addition, the same paper showed that the inflammatory response was significantly more abundant in severe MI mice compared to moderate MI mice, indicating that infarct size is the main determinant for the numerical dynamics of immune cells after MI. Thus, mechanisms different from immune modulation (i.e. the capacity of Tregs to induce myocardial protection and regeneration), might impact on infarct size and eventually also on the extent of inflammation at the site of MI.

We recognize that seminal papers, showing that reperfusion of infarcted cardiac tissue leads to an increased and accelerated inflammatory response, possibly deteriorating cardiac remodeling, date back to the '80s (Entman ML and Smith CW. 1994, *Cardiovasc Res* 28:1301–1311). However, these papers are still highly cited and ample evidence in the literature has shown that overexpression of pro-inflammatory cytokines (i.e. TNF-alpha, IL-1beta, IL-6, MIP-2alpha, MIP-2beta, MIP-3alpha) occurring for more than 7 days after ischemia-reperfusion contribute to cardiac dysfunction and remodeling (Moro et al, Am J Physiol Heart Circ Physiol. 2007 Nov;293(5):H3014-9).

The data presented in our manuscript are consistent with the existing literature, as we also obtained larger infarcts after permanent ligation compared to reperfusion, and the effect of Tregs was evident in both models, consistent with the invariable role of the inflammatory response. Thus, we have embraced the Reviewer's suggestion to rephrase the sentence, also adding two new references on this matter. Nevertheless, we believe that our data are reliable and in line with the existing literature.

The uncritical use of the DEREK mouse (and antibodies) to deplete Tregs is a flawed experiment. Such a depletion leads to a systems wide autoimmune disease and actually kills the mouse after 2-3 weeks because of general inflammation. Thus, the worse post-MI outcome may be unrelated to any influence of Tregs on cardiomyocyte proliferation.

We are aware that Treg depletion has been shown to cause a « catastrophic autoimmunity » (Kim et al. Nat Immunol. 2007 Feb;8(2):191-7). However, this requires an almost complete Treg depletion, whereas the DEREK model used in this work, and even more the antibody-mediated depletion of Tregs, do not deplete Tregs to a level sufficient to lead to autoimmunity. Indeed, the same model has been used by other groups assessing the effect of Treg-depletion in heart pathology (Weirather et al., *Circ Res* 2014;115:55-67).

The tissue analysis also appears subpar and does not match previous reports on the frequency and time course of Tregs after MI (e.g. suppl Fig 3 seems to implicate that Tregs are recruited very early while previous publications using state of the art FACS showed different results).

We believe that the Reviewer refers to 2 studies mentioned before (Yan et al., JMCC 2013, 62:24-35 and Weirather et al., Circ Res 2014;115:55-67). We consider that flow cytometry and immunostaining are complementary methods, with their own limitations. Although we recognize that flow cytometry may lead to more quantitative data, its limitation has been recently shown as *“lymphocyte isolation fails to recover most cells and biases against certain subsets”* (Steinert et al., Cell 2015). Moreover, the work by Weirather et al. showed that Tregs were also present in the remote myocardium. Thus, the apparent discrepancy could be reasonably ascribed to the fact that we limited our analysis to the border region of the myocardial infarction, where CM proliferation was more evident (as our model implies a paracrine effect of Tregs on proliferating CM and therefore requires their close localization). In addition, strain-related differences might additionally explain a delay of a few days in the peak of T cell recruitment, as we have used DREG mice whereas the previous works were performed on C57BL6/J mice.

The in vitro exposure of cardiomyocytes to serum of pregnant mice is interesting, however Tregs do not cross the placenta.

As discussed in our response to Reviewer, although there is no direct proof that maternal Tregs can cross the placenta, there is a well-documented microchimerism between mother and child. Maternal cells populate fetal lymph nodes and have been shown to induce tolerance to maternal alloantigen. As this tolerance is decreased if maternal Tregs are depleted, it has been assumed that maternal Tregs do populate the fetus (J. Mold, Science, 2008). Furthermore, our model implies that Tregs secrete a pool of soluble factors able to promote CM proliferation in a paracrine manner. These Treg-secreted cytokines are able to cross the placenta and reach the fetal heart.

The association of Treg peak in circulation and cardiomyocyte proliferation does not prove causality.

We agree with the Reviewer that our previous data did not prove causality and, following the recommendation of Reviewer 1, we have both assessed CM proliferation in Treg-depleted, pregnant animals, and expanded the discussion on this concept.

Fig. 2E: please improve the unclear group assignment.

We believe that the Reviewer refers to the lack of controls in the previous Figure 2E (now Figure 3E). Representative pictures from control, not pregnant animals, are now included in the picture.

Please add page numbers to your manuscript.

As recommended by the Reviewer, we have added page numbers to the manuscript.

Reviewers' comments:

Reviewer #1 (Remarks to the Author):

This reviewer is positive about the revision and hence acceptance of this work.

The authors answered all the comments of Rev#1 quite successfully. Additionally, rev. #3 was very critical from the immunological point of view, and I liked their answers to him / her).

However, There are still some comments that should be further explained:

1) Point #7: The authors claim that the discrepancies in EF between various experiments reflect variability between strains and individual response to LAD ligation. Regardless of a valid explanation, a significant difference between control and Treg/pools injected groups is clearly observed in Fig. 5D/6D, while the reduction in TD treated group can be hardly observed (Fig. 4C). This may be due to the fact that the initial reduction in EF in both groups (in Fig. 4C) is very mild. I suggest to rephrase the conclusion and state that a mild reduction in LVEF was observed after Treg depletion.

2) According to the reviewer suggestion to add additional proliferation markers, the authors quantified several such markers in Fig. 2. It is quite surprising that the % of EdU+ cells is lower than the rest of the markers, as it is anticipated to be much higher (showing both dividing cells and cells undergoing DNA-synthesis without division). Additionally, the % of AuroraB+ cells should be much lower than the rest, as it reflects genuine cytokinesis. The authors should refer to these differences and try to provide a possible explanation.

Reviewer #2 (Remarks to the Author):

The manuscript by Zacchigna and colleagues has been improved by revision and the inclusion of several new experiments. Most of my concerns have been addressed. I have only a few minor points for the authors' consideration.

Minor Comments:

1. On page 8, the sentence referencing "Figure 3A,B" is mislabeled and corresponds to Figure 4A,B.

2. Depletion of Tregs during myocardial infarction results in lower levels of fibroblast proliferation and less collagen deposition, but larger infarcts (Figure 4). It appears Tregs have a similar pro-proliferative effect on fibroblasts and cardiomyocytes. What are the additional cells that make up the larger infarct with depletion of Tregs? CD45+ cells? Or is the infarct larger but less densely packed with cells? Can the author's comment on this?
3. In Supplementary Figure 2B, please specify in the legend the length of time rat neonatal cardiomyocytes were in culture prior to staining.
4. In Figure 2, please define SN in the legend. Also, are fibroblasts classified as non-alpha-actinin positive cells in the culture? Has a stain for a fibroblast marker been performed to verify these are fibroblasts? Please clarify in the legend.
5. Please clarify the treatment of control animals in Supplementary Figure 4. Are these non-infarcted animals? Sham animals?
6. Supplementary Figure 3 and 6 need statistics for the quantifications.
7. In Figure 1A-B, a no-serum control should be included to indicate the baseline level of proliferation in rat neonatal cardiomyocytes.

Reviewer #3 (Remarks to the Author):

The authors addressed some of my comments (lack of causality, FACS vs histology time course); however, I remain deeply sceptical that the Treg depletion models are appropriate as a lower number of Tregs leads to autoimmune pathologies that likely influence the observed processes.

Reviewers' comments:

Reviewer #1 (Remarks to the Author):

This reviewer is positive about the revision and hence acceptance of this work. The authors answered all the comments of Rev#1 quite successfully. Additionally, rev. #3 was very critical from the immunological point of view, and I liked their answers to him / her).

However, There are still some comments that should be further explained:

1) Point #7: The authors claim that the discrepancies in EF between various experiments reflect variability between strains and individual response to LAD ligation. Regardless of a valid explanation, a significant difference between control and Treg/pools injected groups is clearly observed in Fig. 5D/6D, while the reduction in TD treated group can be hardly observed (Fig. 4C). This may be due to the fact that the initial reduction in EF in both groups (in Fig. 4C) is very mild. I suggest to rephrase the conclusion and state that a mild reduction in LVEF was observed after Treg depletion.

We thank the reviewer for his/her thoughtful suggestion and have rephrased the conclusion.

2) According to the reviewer suggestion to add additional proliferation markers, the authors quantified several such markers in Fig. 2. It is quite surprising that the % of EdU+ cells is lower than the rest of the markers, as it is anticipated to be much higher (showing both dividing cells and cells undergoing DNA-synthesis without division). Additionally, the % of AuroraB+ cells should be much lower than the rest, as it reflects genuine cytokinesis. The authors should refer to these differences and try to provide a possible explanation.

We apologize for this apparent inconsistency, which is now corrected and better explained in the new version of the manuscript. While exposure to EdU lasted for 20 hours in the experiments shown in Figures 1A, B and 6A,B, in the case of Figure 2, EdU was only administered for 10 hours in order to quantify its incorporation also in fibroblasts, which usually proliferate at a higher rate compared to cardiomyocytes. This is the reason for the relatively lower percentage of positive cells. This is now clearly stated in the Methods. In addition, we have replaced the quantification of Aurora B+ cells with the one of AuroraB localization in midbodies, which actually reflects cytokinesis and is indeed observed at a much lower frequency compared to the other proliferation markers.

Reviewer #2 (Remarks to the Author):

The manuscript by Zacchigna and colleagues has been improved by revision and the inclusion of several new experiments. Most of my concerns have been addressed. I have only a few minor points for the authors' consideration.

Minor Comments:

1. On page 8, the sentence referencing "Figure 3A,B" is mislabeled and corresponds to Figure 4A,B.

We apologize for this error, which has been corrected.

2. Depletion of Tregs during myocardial infarction results in lower levels of fibroblast proliferation and less collagen deposition, but larger infarcts (Figure 4). It appears Tregs have a similar pro-proliferative effect on fibroblasts and cardiomyocytes. What are the additional cells that make up the larger infarct with depletion of Tregs? CD45+ cells? Or is the infarct larger but less densely packed with cells? Can the author's comment on this?

As suggested by the Reviewer, our immunofluorescence analysis indicated an increased number of CD45+ inflammatory cells within the scar of depleted mice. Thus, we believe that the large infarct in depleted animals is abundantly composed by inflammatory cells. We have added a sentence in the Discussion, explicitly formulating this hypothesis in the revised version of the manuscript.

3. In Supplementary Figure 2B, please specify in the legend the length of time rat neonatal cardiomyocytes were in culture prior to staining.

As suggested by the Reviewer, we have specified this information in the legend to Supplementary Figure 2.

4. In Figure 2, please define SN in the legend. Also, are fibroblasts classified as non-alpha-actinin positive cells in the culture? Has a stain for a fibroblast marker been performed to verify these are fibroblasts? Please clarify in the legend.

As suggested by the Reviewer, we have defined SN (supernatant) in the legend to Figure 2 and also specified that over the 99% of non-CM cells in our primary cultures are positive for vimentin and therefore classified as fibroblasts.

5. Please clarify the treatment of control animals in Supplementary Figure 4. Are these non-infarcted animals? Sham animals?

As suggested by the Reviewer, we have clarified that control animals in Supplementary Figure 4 are sham operated, non-infarcted.

6. Supplementary Figure 3 and 6 need statistics for the quantifications.

We apologize for this missing information. As the statistical analysis of the data shown in Figures 3 did not show significant differences between E12 and E12 TD groups, we have only indicated the statistical significance relative to NP animals. The analysis of the data shown in Figure 6 did not show any statistical significance and therefore we have added this information in the legend.

7. In Figure 1A-B, a no-serum control should be included to indicate the baseline level of proliferation in rat neonatal cardiomyocytes.

We agree with the Reviewer that a control with no-serum would have been appropriate, but in our experience primary CMs rapidly die in the absence of serum and therefore we could not keep this control up to the end of the experiment.

Reviewer #3 (Remarks to the Author):

The authors addressed some of my comments (lack of causality, FACS vs histology time course); however, I remain deeply sceptical that the Treg depletion models are appropriate as a lower number of Tregs leads to autoimmune pathologies that likely influence the observed processes.

We agree with the Reviewer that the pleiotropic activity of Tregs might introduce some bias in the analysis of the in vivo phenotype. As suggested by both this Reviewer and the Editor, we have now introduced a new set of data in Supplementary Figure 1 showing that Treg depletion did not result in either the up-regulation of inflammatory genes nor the accumulation of CD45+ leukocytes in the embryonic hearts harvested from DEREK mothers depleted with DT injection, compared to embryos harvested from control, non DT-injected, DEREK mothers.

REVIEWERS' COMMENTS:

Reviewer #3 (Remarks to the Author):

I have no further comments.